# $^2$H and $^{18}$O depletion of water close to organic surfaces

Guo Chen, Karl Auerswald, Hans Schnyder

Lehrstuhl für Grünlandlehre, Technische Universität München, Alte Akademie 12, Freising-Weihenstephan 85354, Germany.

*Correspondence to:* Karl Auerswald (auerswald@wzw.tum.de)

**Abstract.** Hydrophilic surfaces influence the structure of water close to them and may thus affect the isotope composition of water. Such an effect should be relevant and detectable for materials with large surface areas and low water contents. The relationship between the volumetric solid:water ratio and the isotopic fractionation between adsorbed water and unconfined water was investigated for the materials silage, hay, organic soil (litter), filter paper, cotton, casein and flour. Each of these materials was equilibrated via the gas phase with unconfined water of known isotopic composition to quantify the isotopic difference between adsorbed water and unconfined water. Across all materials, isotopic fractionation was significant ($p <$ 0.05) and negative (on average -0.91 ± 0.22 ‰ for $^{18/16}$O and -20.6 ± 2.4 ‰ for $^{2/1}$H at an average solid:water ratio of 0.9). The observed isotopic fractionation was not caused by solutes, volatiles or old water because the fractionation did not disappear for washed or oven dried silage, the isotopic fractionation was also found in filter paper and cotton, and the fractionation was independent of the isotopic composition of the unconfined water. Isotopic fractionation became linearly more negative with increasing volumetric solid:water ratio and even exceeded -4 ‰ for $^{18/16}$O and -44 ‰ for $^{2/1}$H. This fractionation behavior could be modeled by assuming two water layers: a thin layer that is in direct contact and influenced by the surface of the solid and a second layer of varying thickness depending on the total moisture content that is in equilibrium with the surrounding vapor. When we applied the model to soil water under grassland, the soil water extracted from 7 cm and 20 cm depth was significantly closer to local meteoric water than without correction for the surface effect. This study has major implications for the interpretation of the isotopic composition of water extracted from organic matter, especially when the volumetric solid:water ratio is larger than 0.5 or for processes occurring at the solid-water interface.

**Key-words:** isotopic fractionation; protein; cellulose; surface effect; O-18; H-2

## 1 Introduction

The $^{18/16}$O and $^{2/1}$H isotope composition of water reflects climate and many processes within the water cycle (Bowen, 2010; Gat, 1996). Changes in the isotope composition of water can either result from the mixing of water with differing isotopic composition or from the change in isotopic composition by fractionation, especially between vapor and liquid. The vapor/liquid fractionation is not only affected by temperature but also by ion hydration (Kakiuchi, 2007). In aqueous solutions, ions change the activities of the isotopologues of water ($H_2O$, HDO, and $H_2^{18}O$) due to their hydration. This, in turn, causes the isotopic fractionation between aqueous solutions and water vapor to differ from the fractionation between pure water and vapor (Kakiuchi, 2007; Stewart and Friedman, 1975). Similar to salt, the surface of hydrophilic materials also interacts with water molecules creating a two-dimensional ice like water layer near the surface and a three dimensional liquid layer far from the surface (Asay and Kim, 2005; Miranda et al., 1998). Additionally, adsorption may cause an energetic difference between water molecules at the surface of solids and the bulk water molecules (Richard et al., 2007). These structural and energetic differences may cause a difference in isotopic composition between these two layers of water. If

existent, such a surface effect should be strongest in materials with large specific surface area and with low water content. There are some indirect hints from studies of plant water uptake from soil, which show that mobile water differs isotopically from immobile water (Brooks et al., 2010; Evaristo et al., 2015; Tang and Feng, 2001) but to the best of our knowledge, such a surface effect has only been directly studied for clay (Oerter et al., 2014) and silica surfaces (Richard et al., 2007). It is not known how large the effect is for organic matter, which are associated with practically all mineral surfaces in the critical zone or form major constituents of other surfaces in the biosphere (Chorover et al., 2007; Nordt et al., 2012; Vazquez-Ortega et al., 2014).

A surface effect may be detected by establishing equilibrium between water adsorbed to a material and air vapor created by unconfined water with known isotope composition in a closed chamber. If there is no surface effect, then the $^{18/16}$O and $^{2/1}$H isotope composition of the adsorbed water and unconfined water should be identical after equilibration. This is because the isotope composition of water under steady conditions is determined by the isotope composition of the water vapor, air humidity, equilibrium fractionation and kinetic fractionation (Helliker and Griffiths, 2007; Welhan and Fritz, 1977). All of these parameters are identical for adsorbed water and unconfined water when they both share the same atmosphere in a closed chamber for a sufficiently long enough time.

We examined the hypothesis that the surfaces of organic materials influence the isotopic composition of adsorbed water and we choose materials of broad relevance. Silage, the product after anaerobic fermentation of fresh forage, is an important feedstuff, which also  delivers water to the animal and thus influences body water composition (Kohn, 1996; Soest, 1994; Wilkinson, 2005) and animal products like milk. Hay has particularly low water content. Organic horizons at the soil surface provide the interface where most vapor and water flows have to pass (Haverd and Cuntz, 2010). More materials like filter paper, cotton, protein powder, and wheat flour were included to identify whether the chemical identity causes or influences the effect. Finally we had to exclude that the effect resulted from artifacts like old water or volatiles and solutes interfering with the isotope measurements (Martín-Gómez et al., 2015; Schmidt et al., 2012; Schultz et al., 2011; West et al., 2011). Silage, which likely is a source of volatiles and solutes in rather large amounts  (e.g., lactic acid, acetic acid, propionic acid, ethanol, and propanol; Porter and Murray, 2001), was also pretreated by washing and heating to remove potentially interfering substances. Water of contrasting isotope composition was used to identify any old water. Finally, we derived a simple prediction model for the effect and demonstrated its versatility in an application case with environmental samples.

## 2 Materials and Methods

We performed three equilibration experiments. Each equilibration experiment involved the exposure of samples to water vapor which originated from unconfined water, followed by cryogenic water extraction from samples and isotope composition measurement. We use $\delta^{18/16}$O and $\delta^{2/1}$H to describe the isotope composition of oxygen ($^{18/16}$O) and hydrogen ($^{2/1}$H) in water (with $\delta^{18/16}$O or $\delta^{2/1}$H $= R_{sample}/R_{standard}-1$, where $R_{sample}$ and $R_{standard}$ denote the ratio of the abundances of heavy and light isotopes in samples following the international SMOW standard).

### 2.1 Preparation of samples

The materials comprised fresh silage, oven dried silage, washed silage, hay, fibric and hemic litter, filter paper, cotton, casein and wheat flour. Silage was also oven-dried to remove all volatiles and it was also washed to remove all solutes. Fibric litter is slightly decomposed organic material on top of the mineral soil derived from plant litter, thus more decomposed than silage but partly still resembling the structure of plant organs.  Hemic litter is strongly decomposed organic material of low fiber content, which has lost the structure of the plant litter but which contains dark brown soluble substances that dye the water extract (Schoeneberger et al., 2012). More pure materials were included to identify whether the chemical identity causes or influences the effect. We used filter paper and cotton to represent pure cellulose, the most common plant

material, commercial wheat flour to represent less pure carbohydrates including branched carbohydrates and commercial casein powder to represent proteins.

The silage and hay were obtained from a farm near Freising and were cut in pieces (4 cm to 8 cm).The silage was stored in a -18 °C deep freezer while the hay was kept in a dark and dry place before use. The hemic and fibric horizons were gathered from a conifer forest near Freising (Germany) from a Haplic Podzol (according to IUSS Working Group WRB, 2014) area and stored in air tight bags in a refrigerator until use. In order to create a relative big range of water content, half of the litter samples were oven dried (16 h for 100 °C) before the equilibration experiment. Filter paper (Rotilabo®-round filters, type 11A, Germany), made of 100 % cellulose, and bleached medical cotton (Paul Hartmann AG, Germany) were prewetted by spraying because the initially dry filter paper and cotton hardly adsorbed any humidity from air. Both materials were then slightly oven dried for different times (ranging from 0 to 60 min) at 50°C before the equilibration experiment to achieve a water content comparable to that of fresh silage and to create a water content gradient. According to the product information, the casein powder (My Supps GmbH, Germany) contained 90 % natural casein and a small amount of carbohydrates while the commercial wheat flour contained 70.9 % carbohydrates, most of which was starch.

## 2.2 Unconfined water

Five isotopically distinct, unconfined waters were used. We term them very heavy, heavy, tap, light and very light waters according to their relative ranking of $\delta^{18/16}O$ and $\delta^{2/1}H$. These waters were produced from deionized water ($\delta^{18/16}O$ = -10 ‰, $\delta^{2/1}H$ = -70 ‰) by means of a rotary evaporator. Very heavy, heavy, light and very light waters had $\delta^{18/16}O$ values of 15, 2, -15 and -22 ‰, and $\delta^{2/1}H$ values of 125, 21, -113 and -160 ‰ respectively with slight deviations between individual experiments.

## 2.3 Set-up of the equilibration procedure

The different materials were individually placed in closed chambers (glass desiccator vessels with a volume of approximate 20 L with drying agent removed) to equilibrate with unconfined water (Fig. 1). In a preliminary experiment, the effectiveness of the chambers' air seal was verified by flushing the containers with $N_2$, followed by monitoring the concentration of $CO_2$ and water vapour inside the vessels. The concentrations after closing the chamber remained constant, which indicated that leaks were negligible. In another preliminary experiment we assessed the development of humidity in the chamber. The humidity reached 100 % within 20 min (half-life 1.8 min) after we put 200 mL of water at bottom of the chamber (Fig. 1), closed it and started the recycling pump (Laboport, Germany). All equilibration experiments lasted for 100 h. Sun et al. (2014) have shown that even for moist samples equilibration is relatively fast (half-life 20 h). A preliminary experiment with silage showed no significant isotope difference ($p > 0.05$ for both H and O) in silage water between 60 and 100 h of equilibration, which implied that 100 h of equilibration were sufficient to achieve equilibrium conditions. Equilibrium conditions also imply that even if there had been condensation within the atmosphere-circulation system, it will not influence the isotope relation between dish water after equilibration and material water because the condensate will also be equilibrated.

In each experiment, 200 mL of (unconfined) water was placed in a glass bowl (15 cm in diameter) on the bottom of the chamber and dishes containing the material samples under focus (about 3 g fresh matter per dish) were placed on a perforated sill in the chamber. We flushed the chamber with nitrogen gas to remove the air vapor and the oxygen to prevent the decay of the samples. After that we immediately closed the chamber and started the recycling pump to ensure homogeneity within the airspace of the chamber. After 100 h of equilibration, samples were quickly removed from the chamber, placed in 12 mL glass vials sealed with a rubber stopper and wrapped with parafilm. The samples were then stored in a -18 °C freezer until water extraction by cryogenic vacuum distillation, as described by Sun et al. (2014). In addition, the

weight of samples was recorded before and after extraction. During equilibration the unconfined water underwent changes
due to the increase of humidity within the chamber (less than 0.3 % of the added water) and exchange with the varying
amount of sample water (up to 10 %). To determine its isotopic composition when in equilibrium with the sample water, we
sampled 1 mL of unconfined water at the end of equilibration and also subjected it to cryogenic vacuum distillation before
measurement.
The extracted water was analyzed with a Cavity Ring Down (CRD) Spectrometer using a L2120 – i Analyzer (Picarro Inc.,
USA). Measurements were repeated until values became stable around a mean. Mean analytical uncertainties quantified as
SD of different replicate measurements for each sample were ±0.06 ‰ for $\delta^{18/16}O$, and ±0.27 ‰ for $\delta^{2/1}H$. Post-processing
correction was made by running the ChemCorrect[TM] v1.2.0 (Picarro Inc.) to exclude the influence of volatiles according to
Martín-Gómez et al. (2015).

**2.4 Experiment A: Influence of materials**

This experiment focused on the fractionation between water in different materials and unconfined water after equilibration.
Dishes containing oven dried silage, hay, oven dried and fresh hemic litter, oven dried and fresh fibric litter, filter paper,
bleached medical cotton, casein powder, or flour were all placed in different chambers for equilibration with unconfined
water to avoid interference of volatiles in different materials. Eight samples for each material that differed in solid:water
ratio were put in one chamber. Some materials (i.e., litter, filter paper, silage) were replicated in different experiments. The
maximum number of samples for one material (silage) was 72. Flour and casein were powders and prone to form dust during
vacuum water extraction. To prohibit this, the opening of vials containing flour and casein powder was covered by parafilm
with tiny holes.

**2.5 Experiment B: Influence of isotopic composition in unconfined water**

This experiment aimed to find evidence that the isotopic fractionation was independent of the isotopic composition of the
unconfined water. This independence will also prove that the isotopic fractionation cannot be caused by old water within the
materials due to insufficient equilibration. Eight samples of oven dried silage in each case were placed into chambers to
equilibrate with five different unconfined waters.

**2.6 Experiment C: Pretreatment of silage**

This experiment investigated the influence of volatiles on the isotope measurement and it assessed the effect of silage solutes
on isotopic fractionation between silage water and vapor. Fresh silage was divided into three groups (8 samples each): The
first group did not undergo any pretreatment. For the second group, about 20 g of silage was immersed in 7 L of deionized
water for about 2 min, stirred during immersion, then taken out using a colander and flushed with distilled water. After that
we squeezed the silage by hand until no water drained off. This washing process was repeated three times. Finally, we
reduced the water content of the washed silage by drying at 80 °C for 40 min. For the third group, silage was oven dried for
16 h at 100 °C to remove water and organic volatiles. These three groups (we call them fresh silage, washed silage and oven
dried silage, respectively, thereafter) were placed in individual chambers and equilibrated with tap water for 100 h.

**2.7 Statistics**

For statistical evaluation we report two-sided 95 % limits of confidence (abbreviated CL) to separate between treatments and
ordinary least squares regression to describe relations between two variables. Measured values were fitted to expected
relations by minimizing the root mean squared error (RMSE). Statistical requirements (normal distribution) were met in all
cases. Significance, even if not explicitly stated, always refers to $p < 0.05$.

## 2.8 Modelling

Conceptually, we assumed water to be part of one of two pools, which are arranged in a shell-like structure around the solid: an inner shell (or layer) which is in immediate contact or close to the surface of the solid and an outer layer that differs in thickness depending on the moisture content or solid:water ratio of the sample. Assuming that the outer layer has the same isotopic composition as the unconfined water once equilibrium was attained and that the inner layer has an isotopic composition that is influenced by the solid, the isotope composition of total adsorbed water ($\delta_T$) was defined as:

$$\delta_T = f_O \times \delta_U + (1 - f_O) \times \delta_S, \qquad (1)$$

where $f_O$ is the fraction of water in the outer layer isotopically identical to the unconfined water, $\delta_U$ and $\delta_S$ are the isotope compositions of unconfined water and water influenced by the surface.

We defined isotopic fractionation ($\varepsilon_S$) between $\delta_S$ and $\delta_U$ as

$$\varepsilon_{S/U} = (\delta_S - \delta_U)/(1000 + \delta_U) \times 1000 \qquad (2)$$

Combining eq. (1) and (2) leads to:

$$\delta_T = (1000 + \varepsilon_{S/U} \cdot f_O)/1000 \cdot \delta_U + \varepsilon_{S/U} \cdot f_O \qquad (3)$$

From this it follows that the apparent isotopic fractionation ($\varepsilon_a$) between the total water in the material and unconfined water is given as:

$$\varepsilon_{T/U} = (\delta_T - \delta_U)/(1000 + \delta_U) \times 1000 = (1 - f_O) \times \varepsilon_{S/U} = f_I \times \varepsilon_{S/U} \qquad (4)$$

The fraction constituted by the inner layer $f_I$ in eq. (4) can be replaced by the ratio between $R_I$, the volumetric ratio of solid:water associated with the layer that is influenced by the surface, and $R_T$, the volumetric solid:water ratio of total adsorbed water:

$$\varepsilon_{T/U} = \varepsilon_{S/U} \times R_T/R_I \qquad (5).$$

Assuming that the size of the inner layer $R_I$ as well as $\varepsilon_{S/U}$ are constant for a certain material, $\varepsilon_{T/U}$ should be related linearly to $R_T$, which is the volumetric solid:water ratio for the total adsorbed water. The solid volume (exclusive voids) can be calculated by knowing the weight and the particle density of the organic matters (casein: 1.43 g/cm$^3$ (Paul and Raj, 1997); silage, hay, litter, filter paper, cotton and flour: 1.5 g/cm$^3$ (Yoshida, et al., 2006)).

In order to exclude that incomplete extraction caused isotopic fractionation, we compared the observed isotopic fractionation with predictions based on a Rayleigh equation (Araguás-Araguás et al., 1995):

$$\varepsilon_{E/T} = (F^{1/\alpha} - F)/(F - 1) \qquad (6)$$

Where $\varepsilon_{E/T}$ is the predicted isotopic fractionation between the incompletely extracted water (subscript E) and total water (T). $F$ stands for fraction of water remaining in the material after the extraction and $\alpha$ stands for isotope fractionation factor (1.0059 and 1.0366 for $^{2/1}$H and $^{18/16}$O at 80 °C extraction temperature, respectively).

## 2.9 Application case

Soil at 7 cm and 20 cm depths and rain water were sampled at the grassland in Grünschwaige Experimental Station, Germany (48°23'N, 11°50'E, pasture #8 in Schnyder et al. (2006); 8.3 % organic matter, 30 % clay, 22 % sand) at biweekly intervals during the growing season (April to November) from 2006 to 2012 and at weekly intervals during the winter season (October to February) in 2015/2016. Soil sampling was always carried out on dry days at midday (between 11 a.m. and 16 p.m.). Two replicates of soil samples were collected on each sampling date. The data were used (i) to examine if there was an offset between soil water and rain water and (ii) whether the offset can be corrected by accounting for the solid:water ratio according to our model. In order to exclude that the offset is caused by soil evaporation, we only use winter season data. During the winter season, evaporation demand was low (average actual evaporation 0.5 mm/d while average precipitation was 1.9 mm/d; German Weather Service, 2016) and evaporation demand should be entirely met by transpiration and intercepted water due to the complete grass cover. Growing season data are only shown for comparison. We had developed the relation between the volumetric solid:water ratio and the isotopic offset only for organic materials.

These materials differed from the soil in so far as they did not contain minerals. Especially for sand it can be expected that it
practically does not absorb water due to its small surface area. Hence, we considered the sand to be inert and did not consider
it in the volumetric solid:water ratio, which in consequence was calculated from (volume of dry soil excluding sand) / soil
moisture volume. The volume of dry soil excluding sand was calculated by dividing its dry weight by particle density of the
organic and mineral components (1.5 $g/cm^3$ and 2.65 $g/cm^3$, respectively; Chesworth, 2008).

## 3 Results

### 3.1 Experiment A: Influence of materials

The apparent isotopic fractionation (*sensu* eq. 4) of $\delta^{18/16}O$ and $\delta^{2/1}H$ was negative and significant ($p < 0.05$) for all materials,
except for $^{18/16}O$ with filter paper and cotton and for $^{2/1}H$ in a few samples of cotton. The volumetric solid:water ratios
differed between materials but also between different samples within the materials providing a wide range. $\delta^{18/16}O$ and $\delta^{2/1}H$
apparent isotopic fractionation decreased significantly with volumetric solid:water ratio over the range of materials. The
decrease was also significant for the different samples within each material (Fig. 2).

### 3.2 Experiment B: Influence of isotopic composition in unconfined water

The isotope composition of absorbed water correlated closely with the unconfined water due to the wide range compared to
the measurement errors ($R^2 = 0.9990$ and $0.9989$ for $^{18/16}O$ and $^{2/1}H$, respectively; Table 1). However, the regressions showed
that the intercept differed significantly ($p < 0.05$) from zero and the slope from one, which indicated that the isotope
composition of adsorbed water was significantly different from that of unconfined water.
Equation (3) predicted a linear relation between $\delta_T$ and $\delta_U$ similar to the linear regressions shown in Table 1. Different to a
regression, however, the slope and the intercept of eq. (3) are not independent but depend on $\varepsilon_{S/U} \times f_O$. To account for this
dependency, the slope and the intercept of the linear equations were estimated by adjusting $\varepsilon_{S/U} \times f_O$ in eq. (3) to minimize
RMSE, while fitting the measured $\delta_T$ and $\delta_U$ values. The optimal fits lead to:
$\delta^{18/16}O_T = (1000 - 1.23)/1000 \cdot \delta^{18/16}O_U - 1.23$
$\delta^{2/1}H_T = (1000 - 22.6)/1000 \cdot \delta^{18/16}O_U - 22.6$ (7)
The $R^2$ between the predictions resulting from the two-layer model and the measurement were similar to that of the linear
regression ($R^2 = 0.9990$ for $^{18/16}O$ and $0.9989$ for $^{2/1}H$), although the model has one degree of freedom less than the
regression. The resulting optimal $\varepsilon_{S/U} \times f_O$ values were -1.23 ‰ for $^{18}O$ and -22.6 ‰ for $^2H$ meaning that the effect was 18
times stronger for $^2H$ than for $^{18}O$.

Equation (5) predicted that the apparent isotopic fractionation changes linearly with the solid:water ratio. This relation was
highly significant ($p < 0.01$) also in the case when waters with very differently isotopic composition were used ($R^2$: 0.7589
and 0.8599 for $^{18/16}O$ and $^{2/1}H$, respectively; Fig. 3). These relations were identical for very heavy, heavy, tap, light and very
light water.

### 3.3 Experiment C: Pretreatment of silage

There was no significant difference between mean gravimetric water contents (based on dry matter) of washed silage (153 %
± 33 %) and fresh silage (128 % ± 10 %) after 100 h equilibration. The water content of oven dried silage did not reach again
the same water content as fresh silage but was significantly lower (81 % ± 13 %). The apparent isotopic fractionation of
washed silage, oven dried silage and fresh silage all decreased with the solid:water ratio (Fig. 4), as already noted in the
experiment with different materials (Fig. 2) or in investigations with unconfined waters of different isotopic composition
(Fig. 3). Washing and oven drying should have removed most solutes and volatiles respectively and thus have created a large

variation in the amount of solutes and volatiles among the treatments. Still, the relationship between apparent isotopic fractionation of all three types of silage and solid:water ratio followed the same line and the areas overlapped each other for the three types of silage (Fig. 4). This implied that neither the volatiles, which possibly could have adulterated the measurements, nor the solutes, which possibly could have influenced water activity in the silage, were the reason of isotopic fractionation. The different treatments, however, separated along the common line due to their differences in water content, which again corroborated the prediction that the apparent isotopic fractionation should linearly change with solid:water ratio.

**3.4 Combining experiments A, B and C**

When combining all experiments with different materials, different pretreatments and different unconfined waters, apparent isotopic fractionation covered a wide range of about 5 ‰ for $^{18/16}$O and 46 ‰ for $^{2/1}$H  (Fig. 5). Even within the same materials, the range was up to 2.5 ‰ for $^{18/16}$O and 25 ‰ for $^{2/1}$H. Apparent isotopic fractionation within materials linearly decreased with the volumetric solid:water ratio.

The isotopic fractionations predicted for Rayleigh fractionation fell far apart the observed isotopic fractionations (Fig. 5). The average deviation between the expected and the observed $^{2/1}$H isotopic fractionation was about 15 ‰. Furthermore, the slope of the relation between the fractionation of $^{2/1}$H and $^{18/16}$O was significantly steeper ($p < 0.05$) for the observed enrichment than the slope predicted for a Rayleigh process. Additionally, the average $^{2/1}$H fractionation of the materials was -20.6 ‰. This net fractionation could be expected for a Rayleigh process if only 80 % of the water would have been extracted while 20 % remained in the sample. This, however, was not the case because subsequent oven-drying did not cause further weight loss.

**3.5 Application**

For the growing season, soil water at 20 cm depth and 7 cm depth showed a distinct deviation from the local meteoric water line (mean deviation for $^{2/1}$H: -8.1 ‰) with a slope almost identical to that of the meteoric water line (regression lines in Fig. 6a). An identical mismatch was detected for the winter season (markers in Fig. 6a) for which confounding effects of evaporation are minimal and summer season.

The deviation between the winter season data and the local meteoric water line correlated significantly ($p < 0.001$) with the solid:water ratio for 7 cm depth but not for 20 cm depth, which varied less in water content. For both depths, the data moved closer to the local meteoric water line when the influence of confined water was removed by applying the general regression with solid:water ratio from Fig. 2 (Fig. 6b). The mean deviation for $^{2/1}$H changed from -8.1 ‰ to 1.0 ‰ for both depths due to this correction.

**4 Discussion**

The extraction of water from solid-water mixtures can be biased by incomplete extraction (Araguás-Araguás et al., 1995) or by the exchange of hydrogen or oxygen from the soil material with water molecules (Meißner et al., 2014). Here we add another confounding effect, which is the inhomogeneous isotopic composition of water above a solid surface. In the following we will discuss (1) whether the observed effect can be due to measuring errors or other reasons than the proposed surface effect, (2) what could be possible reasons of the surface effect, (3) which fields of application will this surface effect likely be important, and (4) which further work related to the surface effect may follow.

**4.1 Excluding other mechanisms than the proposed surface effect**

The study provided clear evidence that the water adsorbed by organic surfaces differed from what would be expected from the isotopic composition of unconfined water and it showed that this deviation became larger with decreasing water content.

Alternative mechanisms leading to an isotopic fractionation other than the proposed surface effect could be (A) volatiles adulterating the measurements; (B) solutes influencing the isotopic composition of adsorbed water; (C) insufficient equilibration time; (D) incomplete extraction of water; (E) metabolically produced water from microorganisms adhering to the materials; (F) exchange of hydrogen and oxygen between the organic matter and the adsorbed water.

(A) The surface effect was largest for flour and casein that do not produce volatiles. Also the filter paper and cotton, which contain no volatiles, had the decreasing trend between apparent isotopic fractionation and solid:water ratio (Fig. 2). Even for silage the influence of volatiles was not evident because washed or oven-dried silage, which should have lost all their volatiles, behaved identical to fresh silage. Also the error in water content caused by not accounting for volatile losses was negligible. Using the correction function by Porter and Murray (2001) to calculate the true water content from the loss of weight, moves the respective data points of silage in Fig. 4 only invisibly (about 0.03 L/L towards right side).

(B) Solutes in water can influence the isotopic fractionation between water and vapor because the energy stage of water molecules bound in the primary hydration sphere of cations and anions differs from that of the remaining bulk water molecules (Kakiuchi, 2007). This effect has been shown for many salts (e.g., KCl, NaCl, $Na_2SO_4$ and $ZnSO_4$). The strength of this effect varies between different ions and may be small (Kakiuchi, 2007; Sofer and Gat, 1975; Stewart and Friedman, 1975). NaCl even does not have a measurable effect on $^{18/16}O$ (O'Neil and Truesdel, 1991). Most of the solutes in our materials were organics for which the effect is unknown. However, this effect must have been small as the washed silage did not show a different pattern in isotopic fractionation compared to fresh silage (Fig. 4). Also the filter paper of analytical grade and bleached cotton that both should not carry any solutes did not show a different pattern.

(C) Insufficient time for equilibration may especially be relevant for silage and litter, which had the highest initial water content. For silage we could show that the apparent isotopic fractionation was independent of the isotopic composition in the unconfined water (Experiment B) despite the wide range of differently labelled unconfined waters (range for $^{18/16}O$: 32 ‰; range for $^{2/1}H$: 285 ‰). However, any old water would have led to a separation in the apparent isotopic fractionation. In contrast, our results were in accordance with the general rule that isotopic fractionation is independent of the isotope composition of the source, which is also underlying eq. (4) and (5). Furthermore, all our experiments used deionized water prepared from tap water except for the experiment with labelled waters for which we can exclude the existence of old water. Our deionized water was similar in isotopic composition to silage water and soil water. The mean $\delta^{18/16}O$ of our water was -10 ‰ while the mean for 52 fresh silage samples analyzed by Sun et al. (2014) was -11 ‰ (SD 3 ‰). A small fraction of old water thus cannot cause the large observed effects.

(D) An incomplete extraction should cause a large error at low moisture content, similar to the general relation between solid:water ratio and isotopic fractionation that we have observed (Fig. 5). However, the predicted isotopic fractionation by incomplete extraction based on a Rayleigh fractionation fell far apart from the observed isotopic fractionation (Fig. 5). In addition, no significant weight difference before and after oven drying of the samples was observed after vacuum extraction. Incomplete extraction is thus an unlikely explanation.

(E) Kreuzer-Martin et al. (2005) found that 10 % of the total water extracted from *Escherichia coli* cells during the log-phase of growth was generated by metabolism from atmospheric oxygen. Thus, intracellular water was distinguishable from extracellular water in $\delta^{18/16}O$. We flushed the chambers with nitrogen gas before equilibration to reduce availability of atmospheric oxygen and minimize microbial growth. For materials like silage dried at 100 °C or filter paper, any significant microbial growth is unlikely. Furthermore, isotopic adulteration caused by microorganisms should have caused $^{18/16}O$ and $^{2/1}H$ deviations in the opposite direction for the very heavy and the very light labeled experiments akin to the experiments by Kreuzer-Martin et al. (2005). In contrast to this $^{18}O$ and $^{2}H$ were always depleted in our experiments regardless of the isotope composition of unconfined water.

(F) Hydrogen bound to oxygen and nitrogen in many organic materials like bitumen, cellulose, chitin, collagen, keratin or wood may exchange isotopically with ambient water hydrogen (Bowen et al., 2005; Schimmelmann, 1991). At room

temperature, this isotopic exchange occurs rapidly in water and an exchange with vapor is even several orders of magnitude
faster (Bowen et al., 2005; Schimmelmann et al., 1993). Such an exchange would influence the adsorbed water but it would
also influence the unconfined water, which is in equilibrium with the adsorbed water but it could not influence the
fractionation between both. The same would apply for an exchange between carbonate oxygen and water oxygen (Savin and
Hsieh, 1998; Zeebe, 2009) although our samples did not contain any carbonate.

### 4.2 Possible reason for the surface effect

The isotopic fractionations became more negative with increasing solid:water ratio and they followed the predictions of eq.
5. This implied that similar isotopic fractionations existed in different materials and that the simple two-layer model
sufficiently described the experimental values. Abundant evidence exists that the properties of water change close to a
surface (Anderson and Low, 1957; Goldsmith and Muir, 1960; Miranda et al., 1998). A hydrogen-bonded ice like network of
water grows up as the relative humidity increases. Above 60 % relative humidity, the liquid water configuration grows on
top of the ice like layer (Asay and Kim, 2005). This transition from a two-dimensional ice-like water to a three-dimensional
water-like layer has been already been shown in several cases (Kendall and Martin, 2005). As we used 100 % relative
humidity in our chamber, both layers should have been present.
The anomalies of water close to a surface appear not to be particularly affected by the detailed chemical nature of the solid
substrates with which the water is in contact. This is referred to as the "paradoxical effect", which describes that –
independent of the nature of the surface – water close to a solid interface is characterized by long-range ordering including
high-pressure ice polymorphs of low energy (Drost-Hansen, 1978). This agrees with our observation that the difference
between materials was small compared to the large variation of the effect caused by a varying solid-water ratio. The small
differences between materials that appear in Fig. 2 may hence only be an effect due to differences between the different
materials in their specific surface area per volume of solid but not due to their chemical nature. The water content of oven
dried silage (81 % ± 13 %) did not reach again the same water content as fresh silage (128 % ± 10 %) but was significantly
lower, which may be because oven drying changes the surface roughness and other structural properties of silage (Tabibi and
Hollenbeck, 1984).
In accordance with our study, Richard et al. (2007) found that water adsorbed in porous silica tubes was depleted in $^2$H
compared to unconfined water and depletion increased with decreasing water quantity as a result of the interplay of
molecular vibrational frequencies and intermolecular H-bonding. This mostly depends on the difference in zero-point energy
between the $^{16/18}$O–$^{1/2}$H bonds, which is compressed at the transition between the bulk liquid and the confined liquid
influenced by the surface (Richard et al., 2007). Our data show, that the effect is much larger for $^{2/1}$H than for $^{18/16}$O and it
practically disappears for $^{18/16}$O when the solid:water ratio decreases below 0.5 (Fig. 5). This may explain why the effect has
been previously described for $^{2/1}$H but not for $^{18/16}$O. Oerter et al. (2014) investigated water adsorbed to clay and also found
isotopic fractionation. They explained this by the negatively charged clay surface, which increases the ionic strength in the
solution close to the clay surface. Ions are known to cause fraction in their hydration sphere (Kakiuchi, 2007; Stewart and
Friedman, 1975). This mechanism could also be active in our samples although the surface charge of most of our samples
(e.g., cellulose) is much smaller than surface charge of clays, and washing, which should have removed most of the solutes,
did not remove the fractionation.

### 4.3 Fields of application

In our experiments we had only examined organic materials while the soil in our application case also contained minerals.
Given the "paradoxical effect" (Drost-Hansen, 1978) and that we had not found any effect of the nature of the organic
materials on the surface effect, the simplest assumption was that there is also no large difference between organic and
mineral surfaces regarding the isotope effect. This seemed reasonable because pure clay with 30 % water content (equivalent

to 0.8 solid:water content) as used by Oerter et al. (2014) created -0.4 ‰ oxygen isotopic fractionation on average. This was close to the predicted apparent isotopic fractionation (-0.7 ‰) for the same solid:water ratio for organic materials. Oerter et al. (2014), however, also manipulated the composition of the solutes, which are known to affect fractionation and which do not allow direct comparison.

The isotopic composition of water in porous samples is usually determined by extracting all water in order to avoid any shift caused by Rayleigh fractionation. Hence, the inner layer close to the surface and the outer layer will be mixed. We could not estimate the thickness of inner layer for our experimental materials. The high-pressure ice polymorphs near surfaces may be one tenth of a micrometer in thickness (Drost-Hansen, 1978) but other effects at the surface-water interface like effects on solute composition extend to a scale of tens of micrometers and in extreme cases up to 0.25 millimeters (Zheng and Pollack, 2003).

 For many processes, especially in the transport of liquid water (e.g., groundwater recharge, stream flow discharge, water uptake by plants) only the outer, mobile layer will be relevant. The extraction of total water will then give a biased estimate of the mobile water. In accordance with our hypothesis, Brooks et al. (2010) even suggested two different soil water worlds to explain their data (mobile water and tightly bound water), which were not identical in terms of isotope composition. They also measured soil water collected in low-tension lysimeters, which represents mobile water, and bulk soil water extracted cryogenically. Bulk soil water was always more depleted in heavy isotopes than lysimeter water collected at the same depth, which was in line with the isotopic fractionation direction observed in our soil case. Tang and Feng (2001) also found isotopic differences between mobile and immobile water in soil and explained this by incomplete replacement of soil water by rainwater. Our laboratory experiments aimed to exclude such an effect.  In our application case we also found a consistent offset between rain water and soil water that cannot result from incomplete replacement of old rain water in soil with new rain water because soil water had an offset from the meteoric water line. Such an offset has been shown for many locations around the world (Brooks et al., 2010; Evaristo et al., 2015), which challenges the assumption in land surface models that plants and streams derive their water from a single, well mixed subsurface water reservoir. Additionally, the surface effect may also play a role in the fractionation between source water and xylem water that has been described for some xerophytic and halophytic species (e.g. Ellsworth and Williams 2007) for which an explanation is presently missing.

In other cases, which focus on the liquid-solid interface, only the water of the inner layer, which is influenced by the surface effect, will be relevant. For example, in studies of cell wall formation or degradation, the total water should be a biased estimate of the isotopic composition near the cell wall. Due to the change in apparent isotopic fractionation with water content, the total cell water will change just by a variation in vacuole volume even if the isotopic composition near the cell wall and in the vacuole remain unchanged. Another example is the determination of fraction of exchangeable hydrogen in organic tissues, which is needed to trace the origin of animals (such as the protein in hair, Bowen et al., 2005). This is usually determined by exposing the tissue to vapor in equilibrium with either heavy or light water similar to our experiments. The surface effect may thus also play a role for the exchangeable hydrogen.

## 4.4 Further work

Solid:water ratio is clearly not the best parameter to describe the two-layer model. The relation should be influenced by specific surface area and by wettability. Hence, the water volume per wetted surface area would likely be a better parameter. For instance, when we wet the filter paper inhomogeneously, we got random results because the average solid:water ratio neither reflected the situation of the wet spots nor that of the dry spots. Also the increasing scatter for solid:water ratios >1.5 (Fig. 5) likely resulted from an inhomogeneous water distribution in these rather dry samples that may have left some parts of the sample completely dry and thus underestimated the water content of other parts.  Still, our model was easy to apply and it worked sufficiently for the wide variety of materials examined. More materials varying in hygroscopic/hydrophobic

behavior and in surface area should be included to better understand the rule behind the variation of isotopic fractionation and to expand the model.

**5 Conclusions**

There was an abundance of evidence to suggest that the surface effect influenced the isotopic fractionation between water adsorbed by organic matter and unconfined water. Many hypothetical reasons for an erroneous isotopic fractionation could be excluded. The variation of apparent isotopic fractionation with water content was well described by a simple, easy to apply two-layer model. This isotopic fractionation should not be neglected when the surface area is huge and the water content is low. The surface effect will become especially relevant for processes happening at the liquid-surface interface like the growth or degradation of the organic materials.

**Author contribution**

G.C. and K.A. designed the experiments and analyzed the data. G.C. carried out the experiments and wrote a first draft. All authors developed and approved the manuscript.

**Acknowledgements**

We gratefully acknowledge financial support from the China Scholarship Council (CSC). We sincerely thank Rudi Schäufele, Wolfgang Feneis and Anja Schmidt for the measurement support and Jason Hartmann for linguistic help.

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

**Table 1: Regressions between water adsorbed by silage ($\delta_T$) and unconfined water ($\delta_U$) for five types of water (very heavy, heavy,**
**tap, light and very light water) based on equation $\delta_T$ = slope × $\delta_U$ + intercept; n = 40; values in parenthesis denote the 95 %**
**confidence level.**

| | $\delta^{18/16}O$ | $\delta^{2/1}H$ |
|---|---|---|
| Intercept | -1.30 (± 0.14) | -22.9 (± 1.1) |
| Slope | 0.987 (± 0.010) | 0.968 (± 0.011) |
| R² | 0.9990 | 0.9989 |




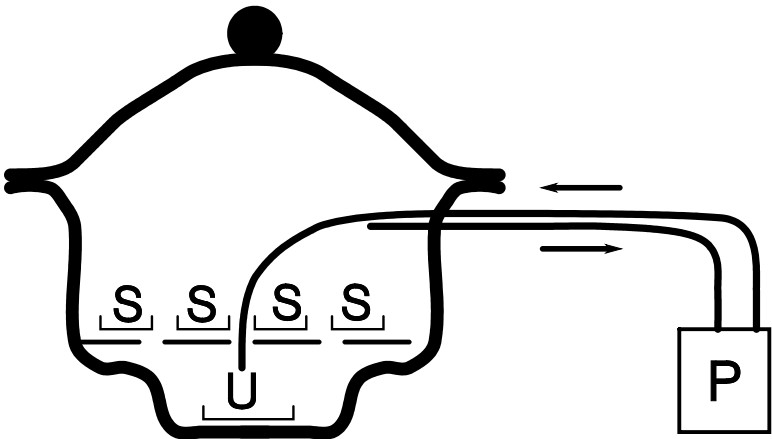


**Fig 1: Experimental set-up with a desiccator vessel as the equilibration chamber. P: recycling pump ensuring air mixing and air**
**movement within the chamber; U: unconfined water filled in the bottom part of the chamber; S: samples placed on top of the**
**perforated middle plate. The arrows indicate the direction of air flow. Vaseline was used as sealant between the lid and the vessel.**




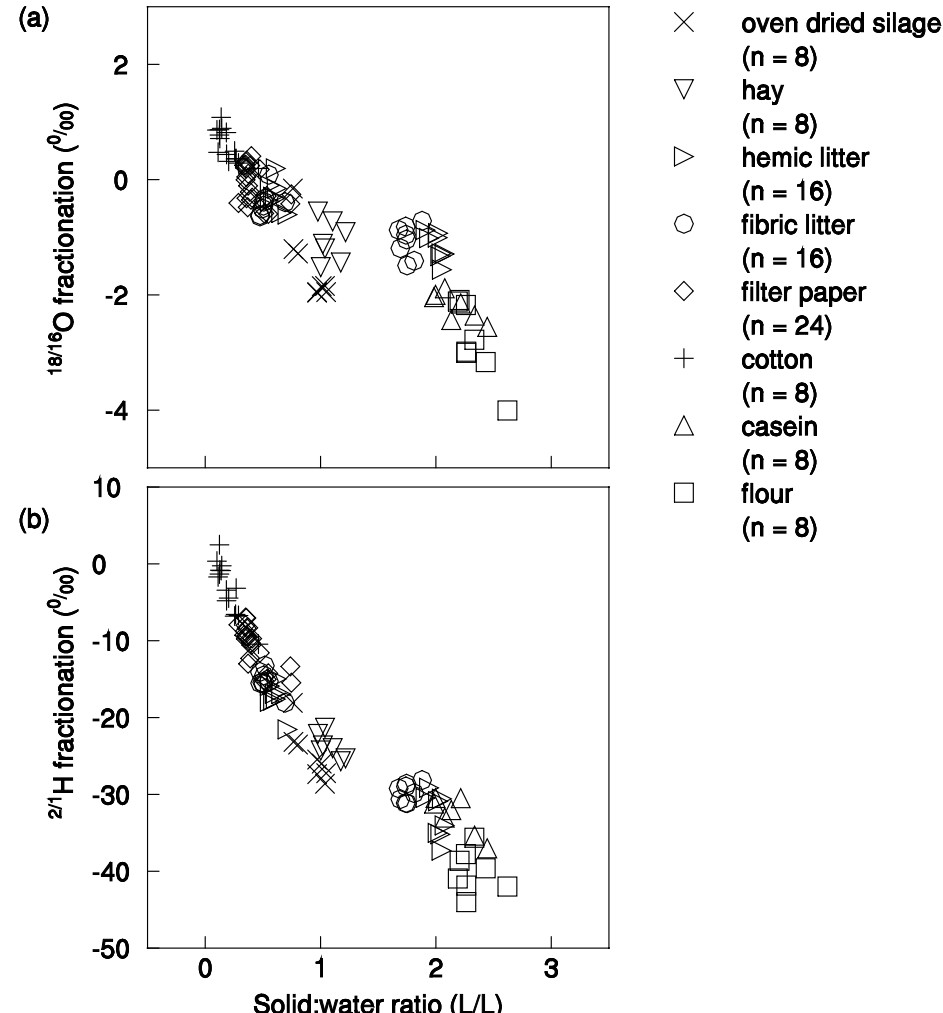

(a)

oven dried silage
(n = 8)

hay
(n = 8)

hemic litter
(n = 16)

fibric litter
(n = 16)

filter paper
(n = 24)

cotton
(n = 8)

casein
(n = 8)

flour
(n = 8)


**Fig. 2: Relationship between volumetric solid:water ratio and apparent isotopic fractionation of (a) $^{18/16}$O and (b) $^{2/1}$H between unconfined water and total water adsorbed by different materials. Taken together, the regressions are y = -0.906 x ($R^2$ = 0.6789; N = 96) for the isotopic fractionation of $^{18/16}$O and y = -17.75 x ($R^2$ = 0.8355) for the isotopic fractionation of $^{2/1}$H.**



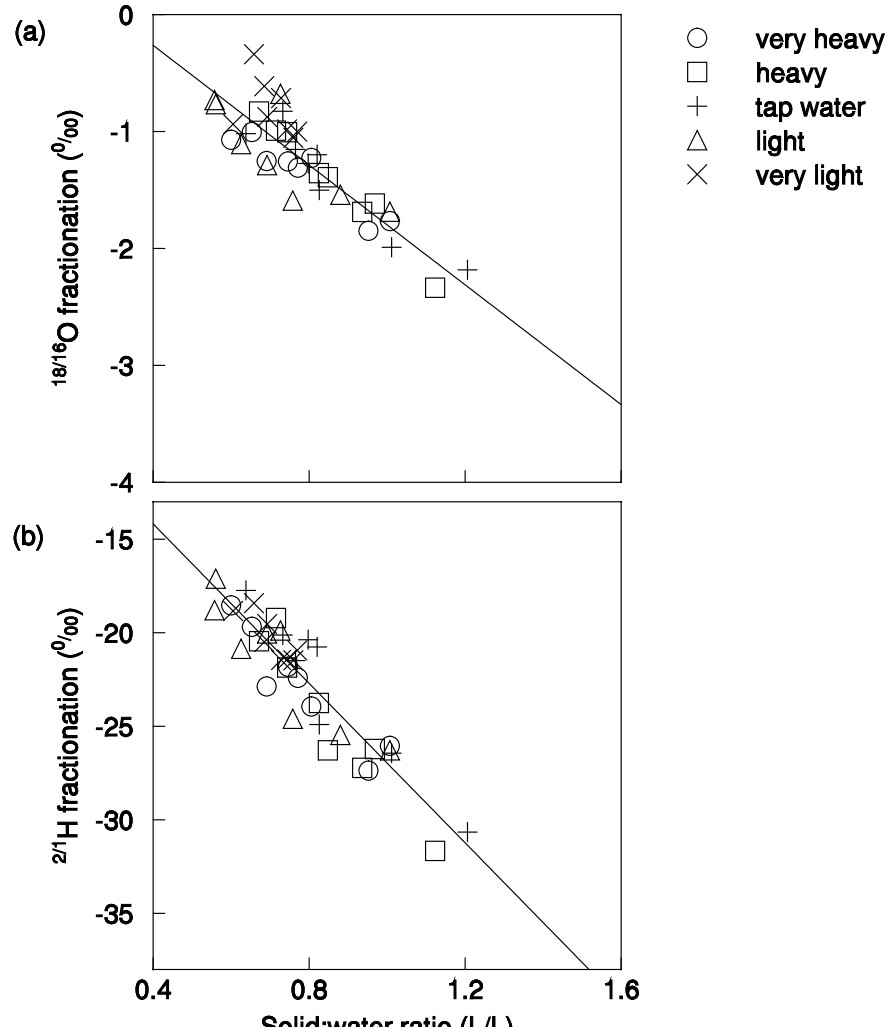

**Fig. 3: Relationship between volumetric solid:water ratio and apparent (a)** $^{18/16}$**O and (b)** $^{2/1}$**H isotopic fractionation of total water**
**absorbed by silage compared to unconfined waters with different isotopic composition. The lines show the best fit (see eq. 7).**

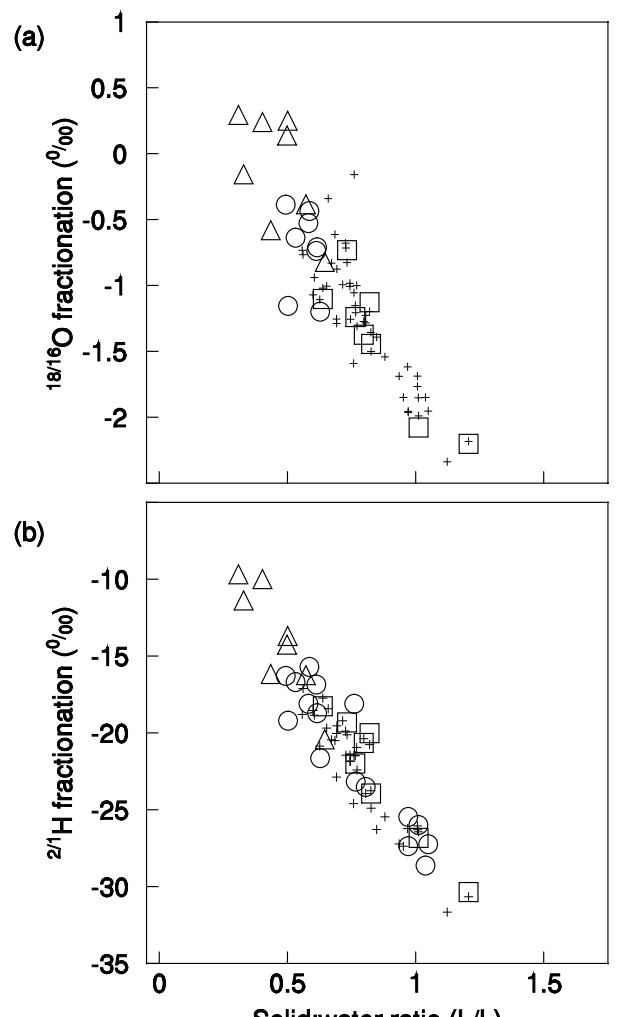


**Fig. 4: Relationship between volumetric solid:water ratio and the apparent isotopic fractionation of (a) $^{18/16}$O and (b) $^{2/1}$H between**
**unconfined water and total water adsorbed by silage with different pretreatments (N = 8 each). The data of Fig. 2 and Fig. 3 (both**
**oven dried silage, N = 32) are provided for comparison.**




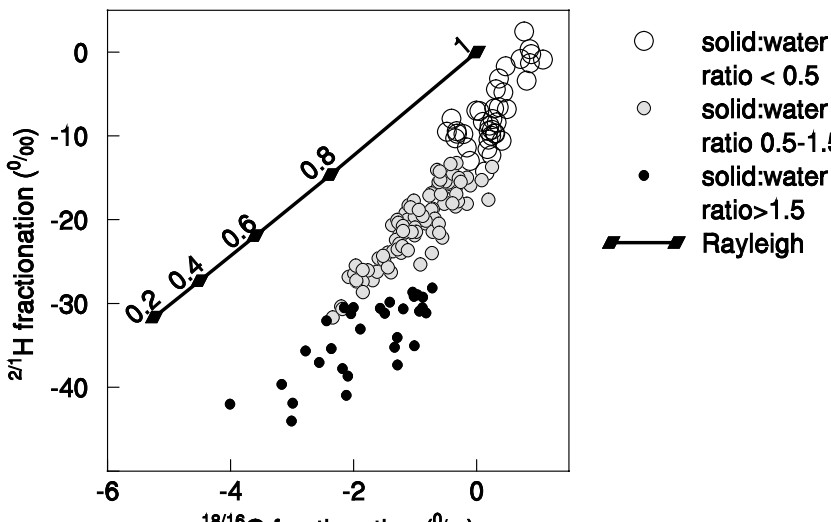


**Fig. 5: Apparent isotopic fractionation (²ˊ¹H versus ¹⁸ˊ¹⁶O) of extracted water as observed in all experiments (markers indicate**
**three groups of solid:water ratios) and fractionation as expected from Rayleigh fractionation (line; numbers denote the fraction of**
**extracted water).**


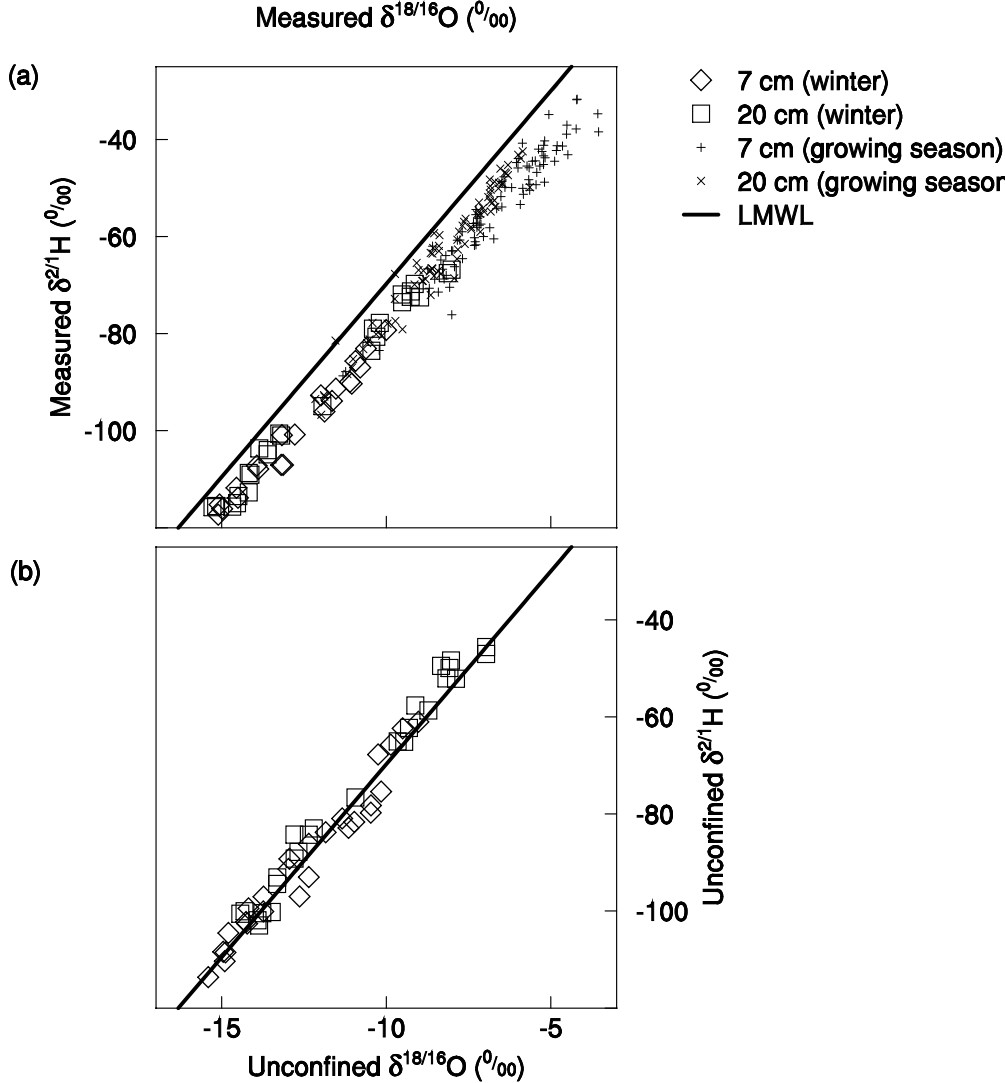

**Fig. 6: Isotope composition of soil water at 7 cm and 20 cm depth (winter season: N = 26; growing season: N = 48). (a) Measured**
**total soil water. (b) Estimated unconfined water. The solid line denotes the local meteoric water line (N = 79; y = (8.0 ± 0.2) x + (10**
**± 2); $R^2$ = 0.99).**