# Peer review of "2H and 18O depletion of water close to organic surfaces"

_Biogeosciences, 2016_

## Referee Comment (RC1) · Anonymous Referee #2 · 28 Mar 2016

Review comments on:

Isotopic offset between unconfined water and water adsorbed to organic matter in equilibrium

by: Guo Chen, Karl Auerswald, Hans Schnyder

General Comments:

This manuscript describes the results of a series of water vapor-adsorbed water experiments conducted to discern isotope fractionation effects related to the surface-water interface. The authors develop a two-compartment model to describe the observations. The results are fairly significant in that they seem to be relevant to a wide range of situations where there is a high soil-water ratio (i.e. soils and vadose sediments).

[Figure]

My main comments are that the authors spend most of the discussion on refuting other explanations for these effects (which is good), but they do not spend much time discussing why these effects are relevant and in what instances. I think the discussion of the natural test case of soils needs to be better discussed and "rounded out" to include how these findings may be used in practice (i.e. What is the use of this effect in casein or flour?). In the abstract, "major implications" are claimed, but they are not ever really discussed. For example, could this effect explain observations of isotopic differences between mobile and immobile water in soils (i.e. Brooks et al., 2010, NatureGeosci, DOI: 10.1038/NGEO722)?

The figures could be improved by denoting each by a specific identifier (e.g. "A" or "B" etc). Figure 6 is relatively unclear to me, and could be improved by a more informative caption. The use of "enrichment" to describe the isotope effect throughout is problematic. Enrichment is used when the isotope effect is both positive and negative. I suggest a more neutral term that is also more descriptive, such as "positive (or negative) isotope effect" to describe the direction of change.

Specific comments by line number:

1. Needs a more specific, or informative title.

13. Avoid the term enrichment. The d values of the water either got higher or lower, or remain unchanged.

14. Define significant

26. modify "discrimination" to something more descriptive and specific to this paper.

31. "Such fractionation can be affected by ion hydration." Needs revising and a reference to support it.

36. "Additionally, adsorption, may cause an energetic difference between water molecules at the surface of solids and the bulk water molecules." – Needs a reference.

[Figure]

51. Please define "silage".

54. Please define "casein powder".

54. What kind (grain type, i.e. wheat?) of flour?

69 – 70. Please briefly define "fibric" and "hemic" rather than sending the reader somewhere else to find a definition that is important to understanding this paper.

78. Why were these materials "slightly dried" before the experiment?

79 -80. How do you know these constituent amounts? Did you do an analysis for this?

80. Why was casein and flour only dried at 50C for 6 hr rather than at 100C and overnight?

99 – 100. Was there condensation inside the pump? How do you know?

104. Was there any evaporative enrichment on the water dish after 100 hr of evaporation?

136. What is OLS?

163. "MIBA Protocol" doesn't mean anything to me. Again, please briefly describe your methods and definitions without making the reader go look somewhere else to understand what you did.

168 "Further, the winter data, effects of soil evaporation from the vegetation covered soil, can be excluded." Please justify and explain why you make this assumption.

171-172. "and sand grains usually are coated by clay, sequioxides, organic matter and biofilms and do not directly interact with water" This seems problematic: If the sand grains are coated with a fractionating substance, how can you neglect them in your analysis? Please address this.

184. Again, define "significantly".

200-201. "The water content of oven dried silage did not reach again the same water content as fresh silage but was significantly lower (81 % $\pm$ 13 %)." This seems important, please address this result.

216-222. Please expand this to include exactly how you calculated the Raleigh fractionation line in Fig. 5.

220-222. "Additionally, an unrealistically small fraction of water would have to be extracted (far below 0.8) to cause the same enrichment of 2 H as observed for most of the samples." Please explain this more.

227: How where the evaporation numbers calculated?

284-289. I do not understand how this explanation rules out exchange of hydrogen and oxygen exchange.

301. Again, you need to explain terms that you invoke, i.e. what is "energy delocalization phenomenon"?

305-311. Oerter et al., 2014 (J. Hydrology) discussed possible reasons for these effects near clay surfaces in some detail. That work should be referenced here.

316. What is the volume ratio of the inner perturbed water to the total bilk water? I assume small, so why would it show up in bulk water extractions?

323-331. In the abstract you claim "major implications" for these results but you do not really discuss any. What are some of the bigger implications of these results?

---

## Referee Comment (RC2) · Anonymous Referee #1 · 31 Mar 2016

**REFEREE COMMENT-DISCUSSION**

Isotopic offset between unconfined water and water adsorbed to organic matter in equilibrium

by G. Chen, K. Auerswald, and H. Schnyder

doi:10.5194/bg-2016-71

**General comments**

In this work, Chen et al. present an experimental study aimed to quantify isotopic fractionation in water adsorbed to organic matter. The work is timely, particularly considering the increasing interest on understanding isotopic fractionation of water within the soil (see e.g. Tang Feng 2001; Brooks et al. 2010). In previous studies, the role

of mineral adsorption and hydration have been explored. However, till now a study on the effect of water adsorption by organic matter was lacking, despite this may also play a significant role in soil processes. In this regard, the present work provides robust experimental evidence of a process with strong implications in different fields, from surface hydrology to the study of plant water uptake.

**Specific comments**

**Experiments**

The methodology of the experiments is generally well described. My main concern is about the calculation of the solid:water ratio. Is it the apparent volume, as typically done for soils? Then, to what extent the porosity of the material could affect the results? For example, depending on how we handle the medical cotton, we can easily modify its volume. Would this change in "solid volume" affect the relationship in the same way as a change in hydration?.

The experiments describe the effect of water-vapour adsorption, but I wonder whether these results could be extrapolable to adsorption processes in the liquid phase, e.g. along the apoplastic water transport in plants. If this were the case, adsorption might explain the fractionation of water between source and xylem water that has been described for some xerophytic and halophytic species (see e.g. Ellsworth Williams 2007). A definitive mechanistic explanation for this process is still lacking and, interestingly, the effect is significant for hydrogen, but not for oxygen. Although right now rather speculative, this topic might deserve consideration for future work.

**Application case**

This section is particularly relevant, as a first attempt to validate the findings of the laboratory experiments. However, the way the data is presented could be improved, and the results deserve more attention in the discussion section.

Firstly, I would expand the methods section, describing the sampling protocol and,
in particular, stating clearly that these values correspond to distilled water from the original soil sampling (i.e. nothing to do with the equilibration experiments).

The authors mention that they did not consider sand for their volume calculations, however, as for the rest of materials, they did not explain how they actually determined the ratio solid:water volume. Is it the apparent volume (i.e. including pores), or an estimate of the solid volume?. In the second case, how was this calculated/estimated? The methodology is likely to be based on standard techniques in soil science, but it is worth to mention them explicitly, particularly to help other researchers to validate the models with their own field data.

On the other hand, the discussion of the application case could be expanded by considering whether the observed relationship can be used to correct field data, and under which conditions. For example, it is worth to discuss why the upper soil seems to fit better with the solid:water ratio than the 20 cm layer. Potentially, this could be related to differences in organic matter content: was this actually the case? On the other hand, the authors apparently pooled together organic matter with clay as "porous" material in their calculations. However, if clay and organic matter do not behave in the same way, this might explain the differences between upper and lower soil layers. Since the soils tested have about 30% of clay, I would try to compare these results with the expected effects of clay minerals, e.g. as in Meissner et al. (2013).

**Technical corrections**

The experiments were designed to test the effect of water-vapour capillary absorption, Adding "vapour" (...water vapour adsorbed...) to the title may help the reader to quickly understand the experimental setup.

Section 2.8 Modelling, and Figures 2-4. In the modelling section, the authors included in the equations the water:solid ratios, whereas in the figures the ratio solid:water is used. I guess that for most readers the water:solid ratio would be more intuitive, so I would suggest to use it also in the figures. Similarly I would replace the term "enrich-
ment" by "offset" or "deviation", which is neutral.

The way the data is presented in Fig.6 is somewhat inconsistent: whereas individual winter values are represented as symbols, the rest of data is presented indirectly with the fitted regression line. I would suggest to present all the data in the same form, ideally as individual dots with their corresponding regression line.

**References**

Brooks,J.R., Barnard,H.R., Coulombe,R. McDonnell,J.J. (2010) Ecohydrologic separation of water between trees and streams in a Mediterranean climate. Nature geoscience, 3, 100-104.

Ellsworth,P.Z. Williams,D.G. (2007) Hydrogen isotope fractionation during water uptake by woody xerophytes. Plant and Soil, 291, 93-107.

Meissner, M., Köhler, M., Schwendenmann, L., Hölscher, D. Dyckmans, J. (2013) Soil water uptake by trees using water stable isotopes (d2H and d8O)-a method test regarding soil moisture, texture and carbonate. Plant and Soil, 376, 327-335.

Tang,K.L. Feng,X.H. (2001) The effect of soil hydrology on the oxygen and hydrogen isotopic compositions of plants' source water. Earth and Planetary Science Letters, 185, 355-367.

---

## Author Response (AR1)

**Dear Editor:**

Below are our point-by-point responses to Ref. #1 (page 1 - 6) and to Ref. #2 (page 7 - 9) and the tracked manuscript showing all our changes (page 10 - end).

**Karl Auerswald**

**Referee #1**

Dear referee. Thank you for your comments. We carefully considered all suggestions. Explanations how we modified the manuscript are given in red below.

My main comments are that the authors spend most of the discussion on refuting other explanations for these effects (which is good), but they do not spend much time discussing why these effects are relevant and in what instances. I think the discussion of the natural test case of soils needs to be better discussed and "rounded out" to include how these findings may be used in practice (i.e. What is the use of this effect in casein or flour?). In the abstract, "major implications" are claimed, but they are not ever really discussed. For example, could this effect explain observations of isotopic differences between mobile and immobile water in soils (i.e. Brooks et al., 2010, NatureGeosci, DOI: 10.1038/NGEO722)?

We reorganized the discussion part and added more relevance in this part and discussed under which condition the findings can be used. We also compared the results of Brooks et al. (2010) with our results.

The figures could be improved by denoting each by a specific identifier (e.g. "A" or "B" etc). Figure 6 is relatively unclear to me, and could be improved by a more informative caption.

We denoted each figure by letters according to your suggestion and added the letter to the corresponding caption. The caption in Fig. 6 was considerably simplified (about 50% shorter).

The use of "enrichment" to describe the isotope effect throughout is problematic. Enrichment is used when the isotope effect is both positive and negative.

I suggest a more neutral term that is also more descriptive, such as "positive (or negative) isotope effect" to describe the direction of change.

Now we use "isotopic fractionation" throughout the manuscript according to the recommendation of Coplen (2011). In order to follow strictly these recommendations we also changed  $\varepsilon_S$  and  $\varepsilon_a$  to  $\varepsilon_{S/U}$ ,  $\varepsilon_{T/U}$  and 18O and 2H were modified to 18/16O and 2/1H.

Coplen, T. B.: Guidelines and recommended terms for expression of stable-isotope-ratio and gas-ratio measurement results, Rapid Commun. Mass Spectrom., 25, 2538-2560, doi: 10.1002/rcm.5129, 2011.

Specific comments by line number:

1. Needs a more specific, or informative title.

We changed the title:

"Isotopic fractionation between unconfined water and water adsorbed in equilibrium to organic matter of biological materials and soils" to

42H and 18O depletion of water close to organic surfaces"

13. Avoid the term enrichment. The d values of the water either got higher or lower, or remain unchanged.

We use 'isotopic fractionation' throughout now (see comment above)

**14. Define significant**

We added *p* values in many cases and additionally wrote in the M&M section "Significance, even if not explicitly stated, always refers to p < 0.05".

26. modify "discrimination" to something more descriptive and specific to this paper. We modified it to isotopic fractionation.

31. "Such fractionation can be affected by ion hydration." Needs revising and a reference to support it.We added a reference and improve the sentence. It now reads:"The vapor/liquid fractionation is not only affected by temperature but also by ion hydration (Kakiuchi, 2007)."

36. "Additionally, adsorption, may cause an energetic difference between water molecules at the surface of solids and the bulk water molecules." – Needs a reference We refer to Richard et al. (2007) now.

Richard et al. 2007 Experimental study of D/H isotopic fractionation factor of water adsorbed on porous silica tubes, Geochim. Cosmochim. Acta, 71.

51. Please define "silage".

We defined silage here. Now it reads:

Silage, the product after anaerobic fermentation of fresh forage, is likely the most important feedstuff in high-productivity ruminant husbandry, which also delivers water to the animal and thus influences the body water composition.

54. Please define "casein powder".

54. What kind (grain type, i.e. wheat?) of flour?

69 - 70. Please briefly define "fibric" and "hemic" rather than sending the reader somewhere else to find a definition that is important to understanding this paper.

Now we explain better the properties of the substances and the rationale behind their selection. In order to make the rationale better visible, we have combined the information, which previously was distributed in the Introduction and in Materials and Methods in one paragraph in M&M. This paragraph reads:

"The materials comprised fresh silage, oven dried silage, washed silage, hay, fibric and hemic litter, filter paper, cotton, casein and wheat flour. Silage was oven-dried to remove all volatiles and it was washed to remove all solutes. Fibric litter is slightly decomposed organic material on top of the mineral soil derived from plant litter, thus more decomposed than silage but partly still resembling the structure of plant organs. Hemic litter is strongly decomposed organic material of low fiber content, which has lost the structure of the plant litter but which contains dark brown soluble substances that dye the water extract (Schoeneberger et al., 2012). More pure materials were included to identify whether the chemical identity causes or influences the effect. We used filter paper and cotton to represent pure cellulose, the most common plant material, commercial wheat flour to represent less pure carbohydrates including branched carbohydrates and casein powder to represent proteins."

Further, we provide the brand name of the casein powder in the following paragraph.

78. Why were these materials "slightly dried" before the experiment?

We now explain the reason like this in manuscript:

"Both materials were then slightly oven dried for different times (ranging from 0 to 60 min) at 50°C before the equilibration experiment to achieve a water content comparable to that of fresh silage and to create a water content gradient."

79 -80. How do you know these constituent amounts? Did you do an analysis for this?

We got the information from the product instruction. Now it reads:

"According to the product information, the casein powder (My Supps GmbH, Germany) contained 90 % natural casein and a small amount of carbohydrates while the commercial wheat flour contained 70.9 % carbohydrates, most of which was starch."

80. Why was casein and flour only dried at 50C for 6 hr rather than at 100C and overnight?

The drying was not a necessary step for these materials. The information was mistakenly introduced here. It was deleted now to avoid confusion.

99 – 100. Was there condensation inside the pump? How do you know?

To be honest, we did not explicitly record whether there was any condensation inside the pump. However, in our experiment we only focused on the final equilibrium after 100 h of exposure, which means that even if there was condensation of water in pump, it will not influence the final equilibrium between the vapor in chamber and dish water (or material water). We improved our explanation:

"A preliminary experiment with silage showed no significant isotope difference (p > 0.05 for both H and O) in silage water between 60 and 100 h of equilibration, which implied that 100 h of equilibration were sufficient to achieve equilibrium conditions. Equilibrium conditions also imply that even if there had been condensation within the atmosphere-circulation system, it will not influence the isotope relation between dish water and material water because the condensate will also be equilibrated."

104. Was there any evaporative enrichment on the water dish after 100 hr of evaporation?

The maximum evaporation is given by the volume of our vessel, temperature and atmospheric pressure. For ambient temperature and atmospheric pressure, the air within the vessel contains about 0.6 g of water. Hence total evaporation was less than 0.3 % of the liquid water within the vessel. We did not examine whether there was any enrichment due to this small evaporative loss because the unconfined water changed its isotopic composition anyway due to the equilibration with the sample water. Measurement errors (e.g. the amount of water added as unconfined water and as sample moisture; isotopic composition of initial and final unconfined water, isotopic composition of sample moisture) were much larger than the expected change due to evaporation. Even this evaporation is irrelevant because we measured the isotopic composition of the unconfined water after the experiment, which included any change due to evaporation that happened after closure of the vessel (100% humidity is reached within 20 min). We modified our description:

"During equilibration the unconfined water underwent changes due to the increase of humidity within the chamber (less than 0.3 % of the total water within the chamber) and exchange with the varying amount of sample water (up to 10 % of the total water). To determine its isotopic composition when in equilibrium with the sample water, we sampled 1 mL unconfined water at the end of equilibration, and also subjected it cryogenic vacuum distillation before measurement."

We deleted the unnecessary abbreviation and write "ordinary least squares" now.

163. "MIBA Protocol" doesn't mean anything to me. Again, please briefly describe your methods and definitions without making the reader go look somewhere else to understand what you did.

We deleted the MIBA protocol and just describe the sampling protocol because the IAEA has deleted this program from their homepage.

168 "Further, the winter data, effects of soil evaporation from the vegetation covered soil, can be excluded." Please justify and explain why you make this assumption.

We apologize. This sentence was a stub. We modified the entire paragraph. See below.

171-172. "and sand grains usually are coated by clay, sequioxides, organic matter and biofilms and do not directly interact with water" This seems problematic: If the sand grains are coated with a fractionating substance, how can you neglect them in your analysis? Please address this.

We modified the entire paragraph:

"The data were used (i) to examine if there was an offset between soil water and rain water and (ii) whether the offset can be corrected by accounting for the solid:water ratio according to our model. In order to exclude that the offset is caused by soil evaporation, we only use winter season data. During the winter season, evaporation demand was low (average actual evaporation 0.5 mm/d while average precipitation was 1.9 mm/d; German Weather Service, 2016) and evaporation demand should be entirely met by transpiration and intercepted water due to the complete grass cover. Growing season data are only shown for comparison. We had developed the relation between the volumetric solid:water ratio and the isotopic offset only for organic materials. These materials differed from the soil in so far as they did not contain minerals. Especially for sand it can be expected that it practically does not absorb water due to its small surface area. Hence, we considered the sand to be inert and did not consider it in the volumetric solid:water ratio, which in consequence was calculated from (volume of dry soil excluding sand) / soil moisture volume. The volume of dry soil excluding sand was calculated by dividing its dry weight by particle density of the organic and mineral components (1.5 g/cm3 and 2.65 g/cm3, respectively; Chesworth, 2008)."

184. Again, define "significantly". We defined it as p < 0.05.

200-201. "The water content of oven dried silage did not reach again the same water content as fresh silage but was significantly lower ( $81 \% \pm 13 \%$ )." This seems important, please address this result.

We explained the possible reasons and added a reference in the discussion part:

"The water content of oven dried silage (81  $\% \pm 13 \%$ ) did not reach again the same water content as fresh silage (128  $\% \pm 10 \%$ ) but was significantly lower, which may be because oven drying changes the surface roughness and other structural properties of silage (Tabibi and Hollenbeck, 1984)."

216-222. Please expand this to include exactly how you calculated the Raleigh fractionation line in Fig. 5.

We explained the Rayleigh fractionation in the M&M section and gave a reference there:

"In order to exclude that incomplete extraction caused isotopic fractionation, we compared the observed isotopic fractionation with predictions based on Rayleigh equation (Araguás-Araguás et al., 1995):

 $\varepsilon_{\rm E/T} = \left(F^{1/\alpha} - F\right) / (F - 1) \tag{7}$

Where  $\varepsilon_{E/T}$  is the predicted isotopic fractionation between the incompletely extracted water (subscript E) and total water (T). *F* stands for fraction of water remaining in the material after the extraction and  $\alpha$  stands for isotope fractionation factor (1.0059 and 1.0366 for H and O at 80 °C extraction temperature, respectively)."

220-222. "Additionally, an unrealistically small fraction of water would have to be extracted (far below 0.8) to cause the same enrichment of  $^{2}$  H as observed for most of the samples." Please explain this more.

We modified the text:

"Additionally, the average 2/1H fractionation of the materials was -20.6 ‰. This net fractionation could be expected for a Rayleigh process if only 80 % of the water would have been extracted while 20 % remained in the sample. This, however, was not the case because subsequent oven-drying did not cause further weight loss."

227: How where the evaporation numbers calculated?

This information was now moved to the M&M section where we provide a reference for the data.

284-289. I do not understand how this explanation rules out exchange of hydrogen and oxygen exchange.

We modified this part to make it clearer:

"Hydrogen bound to oxygen and nitrogen in many organic materials like bitumen, cellulose, chitin, collagen, keratin or wood may exchange isotopically with ambient water hydrogen (Bowen et al., 2005; Schimmelmann, 1991). At room temperature, this isotopic exchange occurs rapidly in water and an exchange with vapor is even several orders of magnitude faster (Bowen et al., 2005; Schimmelmann et al., 1993). Such an exchange would influence the adsorbed water but it would also influence the unconfined water, which is in equilibrium with the adsorbed water but it could not influence the fractionation between both. The same would apply for an exchange between carbonate oxygen and water oxygen (Savin and Hsieh, 1998; Zeebe, 2009) but our samples did not contain any carbonate."

301. Again, you need to explain terms that you invoke, i.e. what is "energy delocalization phenomenon"?

We explain the terms now:

This was taken from the references (as we had indicated). Our original sentence was: "This is referred to as the "paradoxical effect" and is tentatively interpreted in terms of an energy delocalization phenomenon (Drost-Hansen, 1978)."

We changed this to become better comprehensible to:

"This is referred to as the "paradoxical effect", which describes that – independent of the nature of the surface – water close to a solid interface is characterized by long-range ordering including high-pressure ice polymorphs of low energy (Drost-Hansen, 1978)."

305-311. Oerter et al., 2014 (J. Hydrology) discussed possible reasons for these effects near clay surfaces in some detail. That work should be referenced here.

We added the possible reasons proposed by Oerter et al. (2014) here:

"Oerter et al. (2014) investigated water adsorbed to clay and also found isotopic fractionation. They explained this by the negatively charged clay surface, which increases the ionic strength in the solution close to the clay surface. Ions are known to cause fraction in their hydration sphere (Kakiuchi, 2007; Stewart and Friedman, 1975). This mechanism could also be active in our samples although the surface charge of most of our samples (e.g., cellulose) is much smaller than surface charge of clays, and washing, which should have removed most of the solutes, did not remove the fractionation."

316. What is the volume ratio of the inner perturbed water to the total bulk water? I assume small, so why would it show up in bulk water extractions?

We could only speculate about this because our measurements do not allow quantifying this. Hence we did not add anything to our manuscript. However, the effect could still be large even if little water is affected in the case when fractionation is large. Furthermore, the surface itself can be large. For clay minerals, for which the surface is better defined and better known, the surface is often in the range of 500 m2/g. For a rather high solid:water ratio of 1:1 this would mean that 1 mL of water is spread on 500 m2, which could cause a large effect even when fractionation is small. It is thus not unlikely that the perturbed water contributes a significant share to the total water. We write it now:

"We could not estimate the thickness of inner layer for our experimental materials. The high-pressure ice polymorphs near surfaces may be one tenth of a micrometer in thickness (Drost-Hansen, 1978) but other effects at the surface-water interface like effects on solute composition extend to a scale of tens of micrometers and in extreme cases up to 0.25 millimeters (Zheng and Pollack, 2003)."

323-331. In the abstract you claim "major implications" for these results but you do not really discuss any. What are some of the bigger implications of these results?

We added more implications involved in many fields in discussion part: We added comparisons with the results in other studies (such as Oerter, Brooks) and discussed the possible reason for the fractionation between source water and xylem water in halophytic species. We also added more implications in the application part (such as the measurement of exchangeable hydrogen):

"Another example is the determination of fraction of exchangeable hydrogen in organic tissues, which is needed to trace the origin of animals (such as the protein in hair, Bowen et al., 2005). This is usually determined by exposing the tissue to vapor in equilibrium with either heavy or light water similar to our experiments. The surface effect may thus also play a role for the exchangeable hydrogen."

**Referee #2**

Dear referee. Thank you for your comments. We carefully considered all suggestions. Explanations how we modified the manuscript are given in red below.

In this work, Chen et al. present an experimental study aimed to quantify isotopic fractionation in water adsorbed to organic matter. The work is timely, particularly considering the increasing interest on understanding isotopic fractionation of water within the soil (see e.g. Tang Feng 2001; Brooks et al. 2010). In previous studies, the role of mineral adsorption and hydration have been explored. However, till now a study on the effect of water adsorption by organic matter was lacking, despite this may also play a significant role in soil processes. In this regard, the present work provides robust experimental evidence of a process with strong implications in different fields, from surface hydrology to the study of plant water uptake. **Specific comments**

**Experiments**

The methodology of the experiments is generally well described. My main concern is about the calculation of the solid:water ratio. Is it the apparent volume, as typically done for soils? Then, to what extent the porosity of the material could affect the results? For example, depending on how we handle the medical cotton, we can easily modify its volume. Would this change in "solid volume" affect the relationship in the same way as a change in hydration?

The calculation of solid: water ratio is based on the solid volume, which is not the apparent volume. The porosity of the material will not affect this ratio. To make it clearer to the readers, we emphasize that the volume calculation has nothing to do with bulk volume:

"The solid volume (exclusive voids) can be calculated by knowing the weight and the particle density of the organic matters (casein: 1.43 g/cm3 (Paul and Raj, 1997); silage, hay, litter, filter paper, cotton and flour: 1.5 g/cm3 (Yoshida, et al., 2006))."

The experiments describe the effect of water-vapour adsorption, but I wonder whether these results could be extrapolable to adsorption processes in the liquid phase, e.g. along the apoplastic water transport in plants. If this were the case, adsorption might explain the fractionation of water between source and xylem water that has been described for some xerophytic and halophytic species (see e.g. Ellsworth Williams 2007). A definitive mechanistic explanation for this process is still lacking and, interestingly, the effect is significant for hydrogen, but not for oxygen. Although right now rather speculative, this topic might deserve consideration for future work.

We added the discussion about this:

"The surface effect may also play a role in the fractionation between source water and xylem water that has been described for some xerophytic and halophytic species (e.g. Ellsworth and Williams 2007) for which an explanation is presently missing."

**Application case**

This section is particularly relevant, as a first attempt to validate the findings of the laboratory experiments. However, the way the data is presented could be improved, and the results deserve more attention in the discussion section.

Firstly, I would expand the methods section, describing the sampling protocol and, in particular, stating clearly that these values correspond to distilled water from the original soil sampling (i.e. nothing to do with the equilibration experiments).

We expand the methods section and showed our aims of the soil sampling in grassland:

"Soil at 7 cm and 20 cm depths and rain water were sampled at the grassland in Grünschwaige Experimental Station, Germany (48°23'N, 11°50'E, pasture #8 in Schnyder et al. (2006); 8.3 % organic matter, 30 % clay, 22 % sand) at biweekly intervals during the growing season (April to November) from 2006 to 2012 and at weekly intervals during the winter season

(October to February) in 2015/2016. Soil sampling was always carried out on dry days at midday (between 11 a.m. and 16 p.m.). Two replicates of soil samples were collected on each sampling date. The data were used (i) to examine if there was an offset between soil water and rain water and (ii) whether the offset can be corrected by accounting for the solid:water ratio according to our model. In order to exclude that the offset is caused by soil evaporation, we only use winter season data. During the winter season, evaporation demand was low (average actual evaporation 0.5 mm/d while average precipitation was 1.9 mm/d; German Weather Service, 2016) and evaporation demand should be entirely met by transpiration and intercepted water due to the complete grass cover. Growing season data are only shown for comparison."

The authors mention that they did not consider sand for their volume calculations, however, as for the rest of materials, they did not explain how they actually determined the ratio solid:water volume. Is it the apparent volume (i.e. including pores), or an estimate of the solid volume?. In the second case, how was this calculated/estimated? The methodology is likely to be based on standard techniques in soil science, but it is worth to mention them explicitly, particularly to help other researchers to validate the models with their own field data.

The volume of solids, in general, was calculated excluding pores. The volumetric solid:water ratio of soil was estimated by the weight and solid density of the organic matter and minerals without sand. We added the calculation of volumetric solid:water:

"We had developed the relation between the volumetric solid:water ratio and the isotopic offset only for organic materials. These materials differed from the soil in so far as they did not contain minerals. Especially for sand it can be expected that it practically does not absorb water due to its small surface area. Hence, we considered the sand to be inert and did not consider it in the volumetric solid:water ratio, which in consequence was calculated from (volume of dry soil excluding sand) / soil moisture volume. The volume of dry soil excluding sand was calculated by dividing its dry weight by particle density of the organic and mineral components  $(1.5 \text{ g/cm}^3 \text{ and } 2.65 \text{ g/cm}^3, \text{ respectively; Chesworth, 2008}$ ."

On the other hand, the discussion of the application case could be expanded by considering whether the observed relationship can be used to correct field data, and under which conditions. For example, it is worth to discuss why the upper soil seems to fit better with the solid:water ratio than the 20 cm layer. Potentially, this could be related to differences in organic matter content: was this actually the case?

There was a small difference in organic matter content between both depths (the soil was arable about 10 yr before sampling) but we do not expand on this because we do not use this information.

We modified the results part:

"The deviation between the winter season data and the local meteoric water line correlated significantly (p < 0.001) with the solid:water ratio for 7 cm depth but not for 20 cm depth, which varied less in water content. For both depths, the data moved closer to the local meteoric water line when the influence of confined water was removed by applying the general regression with solid:water ratio from Fig. 2 (Fig. 6b). The mean deviation for 2/1H changed from -8.1 ‰ to 1.0 ‰ for both depths due to this correction."

On the other hand, the authors apparently pooled together organic matter with clay as "porous" material in their calculations. However, if clay and organic matter do not behave in the same way, this might explain the differences between upper and lower soil layers. Since the soils tested have about 30% of clay, I would try to compare these results with the expected effects of clay minerals, e.g. as in Meissner et al. (2013).

Meissner et al. used a different experimental setup, in which they did not equilibrate sample water with unconfined water but they compared extracted water with the water added. With this approach they measured the influence of exchange on the

isotopic composition of the extracted water. This is a different process than in our case. Hence we did not compare with the results by Meissner et al. (2013). But we compared our results with the results by Oerter et al. (2014):

"In our experiments we had only examined organic materials while the soil in our application case also contained minerals. Given the "paradoxical effect" (Drost-Hansen, 1978) and that we had not found any effect of the nature of the organic materials on the surface effect, the simplest assumption was that there is also no large difference between organic and mineral surfaces regarding the isotope effect. This seemed reasonable because pure clay with 30 % water content (equivalent to 0.8 solid:water content) as used by Oerter et al. (2014) created -0.4 ‰ oxygen isotopic fractionation on average. This was close to the predicted apparent isotopic fractionation (-0.7 ‰) for the same solid:water ratio for organic materials. Oerter et al. (2014), however, also manipulated the composition of the solutes, which are known to affect fractionation and which do not allow direct comparison. "

**Technical corrections**

The experiments were designed to test the effect of water-vapour capillary absorption, Adding "vapour" (...water vapour adsorbed...) to the title may help the reader to quickly understand the experimental setup.

Actually "vapour adsorption" is not appropriate because most of our materials contained quite a lot of liquid water. We only used a saturated atmosphere for the exchange between unconfined water and material water in order to be able to measure both separately. To improve the title, we changed it:

42H and 18O depletion of water close to organic surfaces".

Section 2.8 Modelling, and Figures 2-4. In the modelling section, the authors included in the equations the water:solid ratios, whereas in the figures the ratio solid:water is used. I guess that for most readers the water:solid ratio would be more intuitive, so I would suggest to use it also in the figures.

Yes, there were inconsistences between the figures and equation in terms of solid:water ratio. It should always read "solid:water ratio".

We agree that the water:solid ratio is much more common but our prediction was that  $\varepsilon_a$  should be related linearly to the volumetric solid:water ratio for the total adsorbed water. A linear equation is easier to fit (e.g., by linear regression) and it would be difficult for the reader so see whether the data really follow an inverse relation and not a deviating curvilinear relationship while it is rather easy to judge a linear relationship.

Similarly I would replace the term "enrichment" by "offset" or "deviation", which is neutral.

We replaced the term "enrichment" by "isotopic fractionation" throughout the manuscript according to your suggestion; we follow strictly Coplen (2011) now.

Reference: Coplen, T. B.: Guidelines and recommended terms for expression of stable-isotope-ratio and gas-ratio measurement results, Rapid Commun. Mass Spectrom., 25, 2538-2560, doi: 10.1002/rcm.5129, 2011.

The way the data is presented in Fig.6 is somewhat inconsistent: whereas individual winter values are represented as symbols, the rest of data is presented indirectly with the fitted regression line. I would suggest to present all the data in the same form, ideally as individual dots with their corresponding regression line.

We made it more consistent: all the values were shown as markers in this figure. However, the summer values (which previously were shown only as lines) are small markers because our arguments are based on the winter data and the summer data a just shown for comparison. And we do not present all regressions in the figure (they would be hard to distinguish because they overlap).

**Isotopic offset between unconfined water and water adsorbed to organic matter in equilibrium2H and 18O depletion of water close to organic surfaces**

4 Guo Chen, Karl Auerswald, Hans Schnyder

5 Lehrstuhl für Grünlandlehre, Technische Universität München, Alte Akademie 12, Freising-Weihenstephan 85354,
6 Germany.

7 Correspondence to: Karl Auerswald (auerswald@wzw.tum.de)

8

9 Abstract. Hydrophilic surfaces influence the structure of water close to them and may thus affect the isotope composition of water. Such an effect should be relevant and detectable for materials with large surface areas and low water contents. The 10 11 relationship between the volumetric solid:water ratio and the isotopic fractionation enrichment of heavy isotopes in between adsorbed water compared withand unconfined water was investigated for the materials silage, hay, organic soil (litter), filter 12 13 paper, cotton, casein and flour. Each of these materials was equilibrated via the gas phase with unconfined water of known 14 isotopic composition to quantify the isotopic difference between adsorbed water and unconfined water. Across all materials, 15 enrichment-isotopic fractionation of the adsorbed water was significant (Confidence interval at 95 % level did not include 0p < 0.05) and negative (on average  $-0.91 \pm 0.22$  % for 18/16O and  $-20.6 \pm 2.4$  % for 2/1H 
[revised manuscript text omitted]

(1)

- $169 \qquad \delta_{\rm T} = f_{\rm O} \times \delta_{\rm U} + (1 f_{\rm O}) \times \delta_{\rm S},$
- where  $f_0$  is the fraction of water in the outer layer isotopically identical to the unconfined water,  $\delta_U$  and  $\delta_S$  are the isotope compositions of unconfined water and water influenced by the surface.

(2)

- 172 We defined enrichment isotopic fractionation ( $\varepsilon_{s}$ ) between  $\delta_{s}$  and  $\delta_{u}$  as
- **173**  $\underline{\mathcal{E}_{S/U}} = (\delta_{S} \delta_{U}) / (1000 + \delta_{U}) \times 1000$
- 174 Combining eq. (1) and (2) leads to:
- 175  $\frac{\delta_{\mathrm{T}}}{\delta_{\mathrm{T}}} \frac{\frac{1000 + c_{\mathrm{S}} f_{\mathrm{O}}}{\delta_{\mathrm{U}}} \delta_{\mathrm{U}} + c_{\mathrm{S}} f_{\mathrm{O}}}{\delta_{\mathrm{O}}}$ (3)
- 176  $\underline{\delta_{\mathrm{T}}} = (1000 + \varepsilon_{\mathrm{S/U}} \cdot f_{\mathrm{O}})/1000 \cdot \delta_{\mathrm{U}} + \varepsilon_{\mathrm{S/U}} \cdot f_{\mathrm{O}}$ (3)
- 177 From this it follows that the apparent enrichment isotopic fractionation ( $\varepsilon_a$ ) between the total water in the material and 178 unconfined water is given as:
- 179  $\underbrace{\underline{\varepsilon}_{T/U}}_{e_{\vec{\pi}}} = (\delta_{T} \delta_{U})/(1000 + \delta_{U}) \times 1000 = (1 f_{O}) \times \underline{\varepsilon}_{S/U} \underline{\varepsilon}_{s} = f_{I} \times \underline{\varepsilon}_{S/U} \underline{\varepsilon}_{s}$ (4)
- 180 The fraction constituted by the inner layer  $f_{\rm I}$  in eq. (4) can be replaced by the ratio between  $R_{\rm I}$ , the volumetric ratio of 181 water:solidsolid:water associated with the layer that is influenced by the surface, and  $R_{\rm T}$ , the volumetric 182 water:solidsolid:water ratio of total adsorbed water:
- 183  $\underline{\varepsilon}_{T/U} \underline{\varepsilon}_{a} = \underline{\varepsilon}_{S/U} \underline{\varepsilon}_{S} \times \underline{R}_{I} / \underline{R}_{T} \underline{R}_{I} / \underline{R}_{I}$ (5).
- Assuming that the size of the inner layer  $R_{\rm I}$  as well as  $\underline{e}_{S/U}e_s$  are constant for a certain material,  $\underline{e}_{T/U}e_a$  should be related linearly to the inverse of  $R_{\rm T}$ , which is the volumetric solid:water ratio for the total adsorbed water. The solid volume (not the bulk volume exclusive voids) can be calculated by knowing the weight and the particle density of the organic matters (casein: 1.43 g/cm3 (Paul and Raj, 1997); silage, hay, litter, filter paper, cotton and flour: 1.5 g/cm3 (Yoshida, et al., 2006)).
- 188 In order to exclude that incomplete extraction caused isotopic fractionation, we compared the observed isotopic fractionation
- 189 with predictions based on a Rayleigh equation (Araguás-Araguás et al., 1995):
- 190  $\underline{\varepsilon}_{\underline{E}/\underline{\Gamma}} = (F^{1/\alpha} F)/(F 1)$  (6)
- 191 Where  $\varepsilon_{E/T}$  is the predicted isotopic fractionation between the incompletely extracted water (subscript E) and total water (T).
- 192 F stands for fraction of water remaining in the material after the extraction and  $\alpha$  stands for isotope fractionation factor
- 193 (1.0059 and 1.0366 for  $^{2/1}$ H and  $^{18/16}$ O at 80 °C extraction temperature, respectively).

**194 2.9 Application case**

Soil at 7 cm and 20 cm depths and rain water were sampled at the grassland in Grünschwaige Experimental Station,
 Germany (48°23'N, 11°50'E, pasture #8 in Schnyder et al. (2006); 8.3 % organic matter, 30 % clay, 22 % sand) following the
 MIBA protocol (Moisture Isotopes in Biosphere and Atmosphere) at weekly biweekly intervals during the vegetation
 growing period season (April to November) from 2006 to 2012 and at weekly intervals during the wintereold season

199 (October to February) in 2015/2016. Soil sampling was always carried out on dry days at midday (between 11 a.m. and 16

- 200 p.m.). Two replicates of soil samples were collected on each sampling date. The data were used (i) to analyzeexamine if 201 there was an offset between soil water and rain water and (ii) whether the offset can be corrected by accounting for the 202 solid:water ratio according to our model. In order to exclude that the offset is caused by soil evaporation, we only use winter 203 season data. During the winter season, evaporation demand was low (average actual evaporation 0.5 mm/d while average 204 precipitation was 1.9 mm/d; German Weather Service, 2016) and evaporation demand should be entirely met by 205 transpiration and intercepted water due to the complete grass cover. Growing season data are only shown for comparison. 206 We had developed the relation between the volumetric solid:water ratio and the isotopic offset only for organic materials. 207 These materials differed from the soil in so far as they did not contain minerals. Especially for sand it can be expected that it 208 practically does not absorb water due to its small surface area. Hence, we considered the sand to be inert and did not consider 209 it in the volumetric solid:water ratio, which in consequence was calculated from (volume of dry soil excluding sand) / soil 210 moisture volume. The volume of dry soil excluding sand was calculated by dividing its dry weight by particle density of the 211 organic and mineral components (1.5 g/cm3 and 2.65 g/cm3, respectively; Chesworth, 2008)-without sand.
- Further, the winter data, effects of soil evaporation from the vegetation covered soil, can be excluded. We verify if the offset can be corrected by accounting for the volumetric solid:water ratio of the soil according to our model. To this end, the sand content of the soil was not considered in the calculation of the solid:water ratio given that the contribution of sand to water storage is marginal (Walczak et al., 2002) and sand grains usually are coated by clay, sequioxides, organic matter and biofilms and do not directly interact with water (Bisdom et al., 1993; Bolster et al., 2001).

**217 **3 Results**

**218 **3.1 Experiment A: Influence of materials**

The apparent enrichment\_isotopic fractionation (sensu eq. 4) of  $\delta^{18/16}$ O and  $\delta^{2/1}$ H was negative and significant (p < 0.05) for all materials, except for 18/16O with filter paper and cotton and for 2/1H in a few samples of cotton. The volumetric solid:water ratios differed between materials but also between different samples within the materials providing a wide range.  $\delta^{18/16}$ O and  $\delta^{2/1}$ H apparent enrichments-isotopic fractionation decreased significantly with volumetric solid:water ratio over the range of materials. The decrease was also significant for the different samples within each material (Fig. 2).

**224 3.2 Experiment B: Influence of isotopic composition in unconfined water**

- The isotope composition of absorbed water correlated closely with the unconfined water due to the wide range compared to the measurement errors ( $R^2 = 0.9990$  and 0.9989 for 18/16O and 2/1H, respectively; Table 1). However, the regressions showed that the intercept differed significantly (p < 0.05) from zero and the slope from one, which indicated that the isotope composition of adsorbed water was significantly different from that of unconfined water.
- Equation (3) predicted a linear relation between  $\delta_{T}$  and  $\delta_{U}$  similar to the linear regressions shown in Table 1. Different to a regression, however, the slope and the intercept of eq. (3) are not independent but depend on  $\varepsilon_{S/U}\varepsilon_{S} \times f_{O}$ . To account for this dependency, the slope and the intercept of the linear equations were estimated by adjusting  $\varepsilon_{S/U}\varepsilon_{S} \times f_{O}$  in eq. (3) to minimize RMSE, while fitting the measured  $\delta_{T}$  and  $\delta_{U}$  values. The optimal fits lead to:

$$\frac{{}^{18}_{--}\Theta}{-} = \frac{\delta_{+-}}{\delta_{+-}} = \frac{1000 - 1.23}{1000} \times \delta_{+-} = \frac{1.23}{1000} \times \delta_{+-} = \frac{1000 - 22.6}{1000} \times \delta_$$

- **234**  $\underline{\delta^{18/16}O_{\mathrm{T}}} = (1000 1.23)/1000 \cdot \underline{\delta^{18/16}O_{\mathrm{U}}} 1.23$
- 235  $\delta^{2/1}H_{\rm T} = (1000 22.6)/1000 \cdot \delta^{18/16}O_{\rm U} 22.6$  (67)
- 236 The  $R^2$  between the predictions resulting from the two-layer model and the measurement were similar to that of the linear
- 237 | regression ( $R^2 = 0.9990$  for  ${}^{18/16}O$  and 0.9989 for  ${}^{2/1}H$ ), although the model has one degree of freedom less than the

- 238 regression. The resulting optimal  $\underline{e_{S/U}} \mathbf{e_s} \times f_0$  values were -1.23 % for 18O and -22.6 % for 2H meaning that the effect was 18 times stronger for 2H than for 18O.
- 240

Equation (5) predicted that the apparent enrichment-isotopic fractionation changes linearly with the solid:water ratio. This relation was highly significant (p < 0.01) also in the case when waters with very differently isotopic composition were used (R2: 0.7589 and 0.8599 for 18/16O and 2/1H, respectively; Fig. 3). These relations were identical for very heavy, heavy, tap, light and very light water.

**245 **3.3 Experiment C: Pretreatment of silage**

246 There was no significant difference between mean gravimetric water contents (based on dry matter) of washed silage (153 % 247  $\pm$  33 %) and fresh silage (128 %  $\pm$  10 %) after 100 h equilibration. The water content of oven dried silage did not reach again 248 the same water content as fresh silage but was significantly lower (81  $\% \pm 13$  %). The apparent enrichment-isotopic 249 fractionation of washed silage, oven dried silage and fresh silage all decreased with the solid:water ratio (Fig. 4), as already 250 noted in the experiment with different materials (Fig. 2) or in investigations with unconfined waters of different isotopic 251 composition (Fig. 3). Washing and oven drying should have removed most solutes and volatiles respectively and thus have 252 created a large variation in the amount of solutes and volatiles among the treatments. Still, the relationship between 253 enrichment apparent isotopic fractionation of all three types of silage and solid:water ratio followed the same line and the 254 areas overlapped each other for the three types of silage (Fig. 4). This implied that neither the volatiles, which possibly could 255 have adulterated the measurements, nor the solutes, which possibly could have influenced water activity in the silage, were 256 the reason of enrichmentisotopic fractionation. The different treatments, however, separated along the common line due to 257 their differences in water content, which again corroborated the prediction that the apparent enrichment isotopic fractionation 258 should linearly change with solid:water ratio.

**259 3.4 Combining experiments A, B and C**

- 260 When combining all experiments with different materials, different pretreatments and different unconfined waters, apparent 261 enrichments isotopic fractionation covered a wide range of about 5 % for  ${}^{18/16}$ O and 46 % for  ${}^{2/1}$ H (Fig. 5). Even within the 262 same materials, the range was up to 2.5 % for  ${}^{18/16}$ O and 25 % for  ${}^{2/1}$ H. Apparent enrichments isotopic fractionation within 263 materials linearly decreased with the volumetric solid:water ratio.
- The enrichments-isotopic fractionations predicted for Rayleigh fractionation fell far apart the observed enrichments isotopic fractionations (Fig. 5). The average deviation between the expected and the observed  $^{2/1}$ H enrichment-isotopic fractionation was about 15 ‰. Furthermore, the slope of the relation between the enrichment-fractionation of  $^{2/1}$ H and  $^{18/16}$ O was significantly steeper (p < 0.05) for the observed enrichment than the slope predicted for a Rayleigh process. Additionally, the average  $^{2/1}$ H fractionation of the materials iswas -20.6 ‰. This net fractionation could be expected for a Rayleigh process if only 80 % of the water would have been extracted while 20 % remained in the sample. This, however, was not the case because subsequent oven-drying did not cause further weight loss. , which means that if the isotopic fractionation was caused by incorrelately extracted weight loss. , which means that if the isotopic fractionation was caused
- 271 by incompletely extraction an unrealistically small fraction of water would have to be extracted (far below 0.8) to cause the
- 272 same enrichment isotopic fractionation of 2/1H as observed for most of the samples

[revised manuscript text omitted]
 - Thus the exchange and the subsequent equilibration 340 with the unconfined water will happen within 100 h Thus the H exchange happened within our experiment time (100 h), 341 which will not influence the final equilibration because the exchange has already stopped after 100 h of exposure in the 342 chamber. Furthermore, an exchange of hydrogen would not explain the observed offset in 48O. Some literatures also reported 343 that there was a fractionation between carbonate oxygen and water Ooxygen (Savin and Hsieh, 1998; Zeebe, 2009) ; 344 however, there was no or small amount of carbonate in our materials. Especially for the casein and flour, the O isotopic 345 fractionation still existed although no carbonate contained 
[revised manuscript text omitted]

- 539 Walczak, R., Rovdan, E., and Witkowska Walczak, B.: Water retention characteristics of peat and sand mixtures,
   540 International Agrophysics, 16, 161-165, 2002.
- Welhan, J. A. and Fritz, P.: Evaporation pan isotopic behavior as an index of isotopic evaporation conditions, Geochim.
  Cosmochim. Acta, 41, 682-686, doi: 10.1016/0016-7037(77)90306-4, 1977.
- 543 West, A. G., Goldsmith, G. R., Matimati, I., and Dawson, T. E.: Spectral analysis software improves confidence in plant and
  544 soil water stable isotope analyses performed by isotope ratio infrared spectroscopy (IRIS), Rapid Commun. Mass
  545 Spectrom., 25, 2268-2274, doi: 10.1002/rcm.5126, 2011.
- 546 Wilkinson, J. M.: Silage, Chalcombe Publications, Lincoln UK, 2005.
- 547 WRB: World reference base for soil resources 2014, FAO Press, Rome, 106-191, 2014.
- 548 Yoshida, A., Miyazaki, T., Ashizuka, M., and Ishida, E.: Bioactivity and mechanical properties of cellulose/carbonate
   549 hydroxyapatite composites prepared in situ through mechanochemical reaction, J. Biomater. Appl., 21, 179-194,
   550 doi: 10.1177/0885328206059796, 2006.
- Zeebe, R. E.: Hydration in solution is critical for stable oxygen isotope fractionation between carbonate ion and water,
   Geochim. Cosmochim. Acta, 73, 5283-5291, doi: 10.1016/j.gca.2009.06.013, 2009.
- Zheng, J. M. and Pollack, G. H.: Long-range forces extending from polymer-gel surfaces, Physical Review E, 68, doi:
   10.1103/PhysRevE.68.031408, 2003.
- 555
- 556
- 557

558 Table 1: Regressions between water adsorbed by silage  $(\delta_T)$  and unconfined water  $(\delta_U)$  for five types of water (very heavy, heavy,

559 tap, light and very light water) based on equation  $\delta_T = \text{slope} \times \delta_U + \text{intercept}; n = 40;$  values in parenthesis denote the 95\_% confidence level.

|                | δ 18/16O | $\delta^{2/1}$ H |
|----------------|----------------------------|------------------|
| Intercept      | -1.30 (± 0.14)             | -22.9 (± 1.1)    |
| Slope          | 0.987 (± 0.010)            | 0.968 (± 0.011)  |
| R 2 | 0.9990                     | 0.9989           |

561

Fig 1: Experimental set-up with an exsiccator vessel as the equilibration chamber. P: recycling pump ensuring air mixing and air movement within the chamber; U: unconfined water filled in the bottom part of the chamber; S: samples placed on top of the perforated middle plate. The arrows indicate the direction of air flow. Vaseline was used as sealant between the lid and the vessel.